# Assessing the Reverse-Engineering Abilities of Large Language Models

## Abstract

Using AI to create autonomous researchers has the potential to accelerate scientific discovery. A prerequisite for this vision is analyzing whether an AI model can identify the underlying structure of a black-box system from its behavior. In this paper, we explore how well a large language model (LLM) learns to identify a black-box function from passively observed versus actively collected data. We investigate the reverse-engineering capabilities of LLMs across three distinct types of black-box systems, each chosen to represent different problem domains where future autonomous AI researchers may have considerable impact: programs, formal languages, and math equations. Through extensive experiments, we show that LLMs fail to extract information from observations, reaching a performance plateau that falls short of the Bayesian inference ideal. However, we demonstrate that prompting LLMs to not only observe but also intervene—actively querying the black-box with specific inputs to observe the resulting output—improves performance by allowing LLMs to test edge cases and refine their beliefs. By providing the intervention data from one LLM to another, we show that this improvement is partially tethered to the process of generating effective interventions, paralleling results in the literature on human learning. Further analysis reveals that engaging in interventions can help LLMs escape from two common failure modes: *overcomplication*, where the LLM falsely assumes prior knowledge about the black-box, and *overlooking*, where the LLM fails to incorporate observations. These insights provide practical guidance for helping LLMs more effectively reverse-engineer black-box systems, supporting their use in making new discoveries.

## 1 Introduction

Developing intelligent systems to accelerate scientific discovery has been a long-standing goal of artificial intelligence research (Gil et al., 2014; Wang et al., 2023). Despite rapid progress in creating large language models (LLMs) for understanding text and solving problems in math and coding, automating science poses a different kind of challenge. A core aspect of scientific discovery is *reverse-engineering* the mechanism behind a black-box system, which requires capabilities beyond responding to a one-off query. In particular, reverse-engineering often involves 1) understanding observed data in order to develop hypotheses, 2) designing experiments to actively acquire informative data from the black-box to test those hypotheses, and 3) describing and communicating the results.

Existing work using LLMs for automating scientific processes either focuses on static observational data (Rmus et al., 2025; Shojaee et al., 2025) or emulates scientific workflows using "LLM scientists" with many moving parts (Gandhi et al., 2025; Schmidgall et al., 2025). In contrast, research in related fields has used carefully controlled tasks to evaluate whether machine learning systems can perform key aspects of reverse-engineering, including inductive reasoning (Rule et al., 2024), learning causal features from passive data (Lampinen et al., 2023), and optimal experimental design (Chaloner & Verdinelli, 1995; Rainforth et al., 2024). This work is often informed by work in cognitive science, which has studied how humans engage in active learning using methods in which the source (either passive observation or active experimentation) and content of data can be differentiated (Markant & Gureckis, 2010; 2014). Such controlled methods have not been applied to state-of-the-art LLMs, leaving fundamental questions unanswered: "*How well can LLMs make inferences from passive observations?*" and "*Can they actively collect data to refine their hypotheses?*".

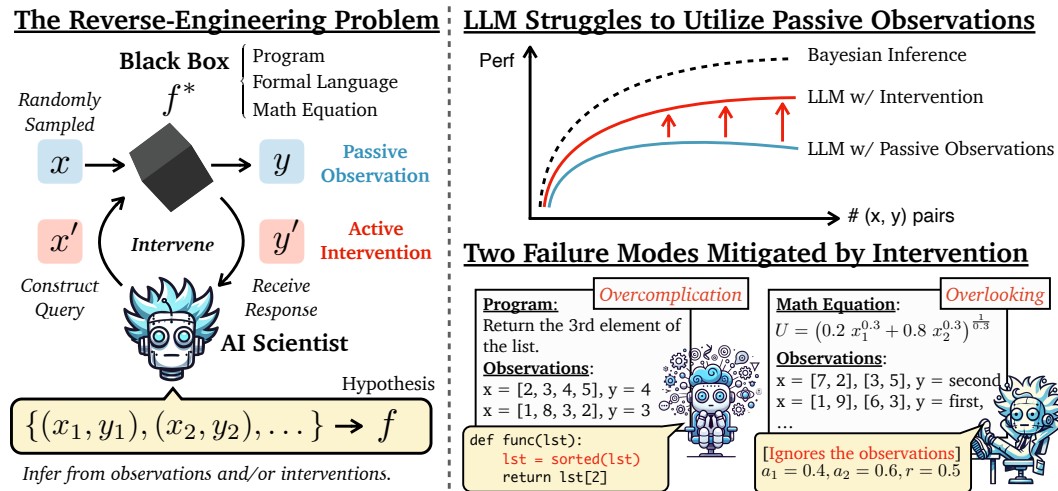

Figure 1: **Reverse-engineering tasks.** Left: Definition of the problem. The AI scientist will obtain passive observations from the black box or collect data through active intervention to construct a hypothesis. Right (top): with only passive observations, the LLM cannot make effective use of the data and lags behind Bayesian inference; allowing the LLM to intervene improves performance. Right (bottom): effective intervention can mitigate two common failure modes: (1) overcomplication—the program blackbox expects returning the 3rd element from a list, and the LM adds an unnecessary sorting step that are not supported by the observations, and (2) overlooking—the math blackbox expects choosing the higher utility list, and the LM predicts the equation parameters that are obviously incompatible with the observed data.

To answer these questions, we systematically study LLMs on three reverse-engineering tasks inspired by the cognitive-science literature and selected to mimic challenges in scientific settings: reconstructing list-mapping programs (Rule et al., 2024), formal languages (McCoy & Griffiths, 2023), and math equations (Foster et al., 2019). Through extensive experiments, we show that LLMs are limited in their ability to make inferences from observations, leading to performance plateaus compared to Bayesian models. However, allowing LLMs to perform interventions—generating test cases or queries to collect new, informative data—can significantly improve their performance.

Through further experiments where outcomes of interventions conducted by one LLM become observational data for another, we show that the benefits of intervention seem to come from the LLM testing and refining its own beliefs rather than simply collecting higher-quality data. This is similar to a phenomenon observed in human learning, where people show limited benefit from interventions generated by others (Markant & Gureckis, 2010; 2014). Further investigation reveals that generating interventions seems to help LLMs overcome two failure modes: 1) *overcomplication*, where the LLM tends to construct overly-complex hypotheses, and 2) *overlooking*, where the LLM neglects observations or draws overly-generic conclusions without careful checking.

Our contributions are as follows:

- Drawing inspiration from controlled studies of human cognition, we formalize *reverse-engineering* — generating hypotheses from observed data and collecting informative data through interventions, as a core problem for assessing the scientific discovery capabilities of LLMs and design three black-box tasks to facilitate such an assessment.

- We demonstrate empirically that frontier LLMs still struggle, relative to Bayesian inference, at reverse-engineering these black boxes from only passive observations.

- We show that LLMs can perform interventions to obtain more informative data, and that effective intervention mitigates the failure modes of *overcomplication* and *overlooking*.

- We show that performance degrades when repurposing the LLM's intervention data as observations, pinpointing the mechanism behind the improvements it produces and highlighting a potential pitfall for exchanging knowledge among LLMs.

## 2 RELATED WORK

**Inductive Inference**   Some of the earliest work on reverse-engineering appears under the label of *inductive inference* for "hypothesizing a general rule from examples" (Angluin & Smith, 1983). Classic instances of this problem include work on identifying the underlying structure of a finite-state automaton through observations of its input-output behavior (Rivest & Schapire, 1987; 1989). While this problem typically considers passive observations, seminal work on active learning focuses on analyzing the benefits of actively querying inputs to solicit the most-informative outputs from the unknown function of interest (Littlestone, 1988; Angluin, 1988; Settles, 2009). The key distinction between these seminal works and ours is the attention towards LLMs and assessing their capacity for successfully identifying different types of black boxes from input-output examples.

**Bayesian Optimal Experiment Design**   An adjacent line of work considers the sequential design of experiments which maximally yield information gain about an unknown parameter of interest (Lindley, 1956; DeGroot, 1962; Chaloner & Verdinelli, 1995; Rainforth et al., 2024); one may interpret these methods as studying a non-LLM-focused, Bayesian analogue of the reverse-engineering problem we formulate in the subsequent section, where a learner begins with a prior distribution over the black box in question and must maximally reduce epistemic uncertainty (Der Kiureghian & Ditlevsen, 2009) with a given budget of experiments. To the extent that LLMs may implicitly engage with an underlying approximate posterior inference scheme (Xie et al., 2021; Griffiths et al., 2024; Zhu & Griffiths, 2024a; Falck et al., 2024; McCoy et al., 2024), the reverse-engineering capabilities studied in this work can be tied to this Bayesian optimal experiment design problem.

**Reinforcement Learning**   The fundamentals of the reverse-engineering problem also connect with various ideas studied in the context of reinforcement learning (RL) (Sutton & Barto, 1998). Any model-based RL agent (Sutton, 1990; 1991; Brafman & Tennenholtz, 2002; Strehl & Littman, 2008) naturally engages with a particular instance of the reverse-engineering problem where the black-box function in question is the transition function and/or reward function of a Markov Decision Process (MDP) (Bellman, 1957; Puterman, 1994). The distinction explored in this work between a LLM that passively observes versus actively intervenes on the black box in question has a direct connection to the exploration challenge in RL, which has profound impact on an agent's ability to recover an accurate model of the world (Thrun & Möller, 1991; Deisenroth & Rasmussen, 2011; Strens, 2000; Osband et al., 2013); while recent work (Arumugam & Griffiths, 2025) has studied how to improve exploration with LLMs, this paper focuses on assessing the innate capabilities of LLMs to actively query informative data. Ostrovski et al. (2021) demonstrate the ineffectiveness of passive learning with deep RL agents and their need to intervene so as to correct misunderstandings about the world; our work provides the LLM complement to their findings. The KWIK learning framework of Li et al. (2008) provides a theoretical analysis for reverse-engineering a MDP transition function when a learner must either confidently estimate the environment dynamics or say "I don't know" (Walsh et al., 2009; Li & Littman, 2010; Sayedi et al., 2010; Szita & Szepesvári, 2011; Abernethy et al., 2013). Finally, there is a connection between intervention for effective reverse-engineering and meta RL (Liu et al., 2021), with recent work showing that passive learning can be effective with LLMs once there is an effective exploration strategy capable of yielding high-quality observations (Lampinen et al., 2023); naturally, the latter problem is precisely what we demonstrate interventions allow LLMs to solve for themselves in reverse-engineering tasks.

**LLMs for Automating the Scientific Process**   With the rapid advances in LLMs, recent work has explored using them to automate different parts of the scientific process such as ideation (Si et al., 2024), assistance (Gottweis et al., 2025), writing research papers (Lu et al., 2024; Starace et al., 2025), or emulating AI scientists in simulated environments (Schmidgall et al., 2025). Complementary lines of research further examine how LLMs capture human inductive biases (Si et al., 2023), elicit human preferences (Li et al., 2023), characterize distributional differences through language (Zhong et al., 2023), and perform Bayesian preference elicitation (Handa et al., 2024), broadening the scope of how LLMs can support scientific reasoning. Additionally, multi-modal and multi-agent AI models have driven significant progress in applications such as protein science (O'Neill et al., 2025), while frameworks like MatPolit (Ni et al., 2024) integrate human cognitive insights to accelerate discoveries in materials science. These works utilize the abundant knowledge stored in LLMs to directly tackle real-world complexity in science (Reddy & Shojaee, 2025). Recent work also

shows that LLMs can autonomously generate and test hypotheses to advance automated scientific discovery (Agarwal et al., 2025). However, the complexity of these settings and the resulting agents make it hard to disentangle the consequences of the engineering choices that go into these systems. Our work focuses on simple and controllable black boxes to study the core capabilities of the LLMs themselves.

**Understanding Failure Modes in LLMs** Recently, many works have examined the failure modes (Aggarwal et al., 2025) of formal reasoning in LLMs. It has been observed that LLMs can exhibit failure modes of both "overthinking" (Chen et al., 2024) and "underthinking" (Wang et al., 2025) when tackling mathematical problems and code generation (He et al., 2025; Sprague et al., 2024) and (Cuadron et al., 2025; Sprague et al., 2024; Sui et al., 2025; Cemri et al., 2025). To understand LLM abilities beyond formal reasoning tasks, recent work has leveraged insights and datasets from cognitive science (Frank, 2023; Binz & Schulz, 2023; Coda-Forno et al., 2024; Ying et al., 2025). In particular, researchers have started to use cognitive science to explore the failed behaviors in LLMs (Ku et al., 2025). Using these methods, researchers have found that LLMs sometimes overestimate human rationality (Liu et al., 2024a), exhibit inconsistencies in probability judgments (Zhu & Griffiths, 2024b), and perform worse as a result of engaging in reasoning (Liu et al., 2024b). In a similar vein, our work draws upon cognitive science to design the black boxes used in our reverse-engineering experiments.

## 3 REVERSE ENGINEERING

### 3.1 PROBLEM FORMULATION

We define a black box $f^* : \mathcal{X} \to \mathcal{Y}$ as a deterministic function that maps a query $x \in \mathcal{X}$ to a response $y \in \mathcal{Y}$ through its internal dynamics. The **reverse-engineering** problem is for a model to infer the internals of a black box $f^*$ (such as list mapping programs, production rules of formal languages, and math equations) from a sequence of query-response pairs $\mathcal{O} = \{(x_1, y_1), (x_2, y_2), \ldots, (x_N, y_N)\} \subset \mathcal{X} \times \mathcal{Y}$ (Figure 1). We consider two cases of the reverse-engineering problem: **observation-only** and **observation-intervention**. In the observation-only scenario, all the queries are *randomly sampled* from $\mathcal{X}$ and the corresponding response $y_i = f^*(x_i)$ is generated by the black box from a uniform distribution to construct the observation set. A large language model $\mathcal{M}$ must generate a hypothesis $f = \mathcal{M}(\mathcal{O})$ without further interaction with the black box. This setting assesses the model's ability to perform inductive reasoning (Angluin & Smith, 1983). In the observation-intervention scenario, the LLM is first given a set of observations $\mathcal{O}$ obtained in the observation-only scenario and is instructed to interact with the black box in a multi-round fashion. In each round, the LLM chooses one of the following actions: 1) construct a new query $x_{N+1}$ to query the black box and obtain the response $y_{N+1}$, 2) construct a new query-response pair $(x_{N+1}, y'_{N+1})$ and check its validity using the black box ($\mathbb{1}[y'_{N+1} = f^*(x_{N+1})]$), or 3) stop and conclude with a hypothesis $f$ about the black box. Before constructing the new query, the LLM can analyze the current observations. We also compare the intervention with strategies such as verbalizing its current belief or describing the current hypothesis (§5.2) and find that "Analyze-then-Query intervention" is the best intervention strategy. Therefore, we adopt Analyze-then-Query as the main intervention strategy for all core experiments.. Before the LLM chooses to stop or reaches the maximal number of rounds, the query-response pairs obtained during intervention are appended to $\mathcal{O}$ for the next round.

### 3.2 BLACK-BOX TYPES

Drawing on the literature on inductive inference in cognitive science, we select tasks commonly used to study learning of complex relationships to design our black-box systems and scale them up for evaluation with LLMs. These three distinct black-box function classes – Program, Formal Language, and Mathematical Equation – simulate problems encountered in scientific reverse-engineering scenarios. Due to space constraints, detailed black-box designs are relegated to Appendix B.

**Program.** We use list-mapping programs (Rule et al., 2024) for the Program black-box. Each program implements a lambda expression (e.g., `(lambda(singleton(third $0))))` in Python, where the query is a list of integers and the response is an integer.

**Formal Language.** The Formal Language black-box is defined by a simple program that generates sequences of symbols. For example, the language $A^n B^n$ generates sequences consisting of some number of $A$s followed by the same number of $B$s. The black-box allows the LLM to intervene by validating if a string is allowable under the rule. We define 46 distinct black boxes each based on a language from Yang & Piantadosi (2022) or McCoy & Griffiths (2023).

**Math Equation.** We use the Constant Elasticity of Substitution (CES) formulation from economics (Foster et al., 2019) as the Math Equation black-box. The utility $U = (\sum_i a_i x_i^r)^{\frac{1}{r}}$ is given by the weights $a_i$, the ratio $r$, and the quantities of each kind of goods $x_i$. The LLM queries the black box with two lists of item types with quantities. The response says which list has higher utility.

### 3.3 EVALUATION PROTOCOL

A black-box can be represented in multiple ways, rendering evaluation challenging. For example, two black-boxes can be compared through their descriptions in natural language (descriptive evaluation) or whether they respond similarly to the same queries (functional evaluation; see §J). In this paper we focus on **descriptive evaluation**, where the black-box $f_{\text{NL}}^*$ is expressed in natural language, due to its communicative nature and real-world use (Chopra et al., 2019; Gandhi et al., 2025). The LLM-generated hypothesis $f_{\text{NL}}$ is scored by an LLM judge against the black-box on a $0 - 10$ scale based on the criteria of each black-box type ($\texttt{score} = \texttt{LM-Judge}(f_{\text{NL}}, f_{\text{NL}}^*)$). We use descriptive evaluation for Program and Formal Language. As the Math Equation does not require verbalization beyond the weights and ratio, we report the flipped root mean square error (1 - RMSE) between the inferred parameters and ground truth. We provide clear rubrics and an example to explain the descriptive evaluation in Appendix E and F.1.

## 4 EXPERIMENTS

**Experimental setup.** We use different versions of GPT-4o (Hurst et al., 2024) for reverse-engineering (*gpt-4o-2024-08-06*, dubbed as reverse-engineer LLM) and as the judge (*gpt-4o-2024-05-13*, dubbed as the judge LLM). We use greedy decoding of both the reverse-engineer and the judge LLMs and report performance over 3 seeds. For the observation-only experiments, we report performance for number of observations $N = \{2, 10, 15, 20, 30, 60\}$. For the observation-intervention setting, the reverse-engineer LLM performs $M = \{5, 10, 20, 50\}$ rounds of interventions conditioned on the initial set of 10 observations ($|\mathcal{O}| = 10$). In addition to GPT-4o, we report full results for Claude-3.5-Sonnet-20241022 (Anthropic, 2024), DeepSeek-R1 (Guo et al., 2025), Llama-3.3-70B-Instruct Grattafiori et al. (2024), GPT-5, and Claude 4 Sonnet in Appendix 5.4, showing that even the strongest models still require active intervention to achieve high performance. We also show the reliability of using GPT-4o as a judge in Appendix I, comparing Cohen's scores with human annotations and with the latest LM judge results. We provide prompts for both intervention and hypothesis generation in Appendix E and other evaluation approaches in Appendix J.

### 4.1 LLM STRUGGLES TO UTILIZE OBSERVATIONS OPTIMALLY

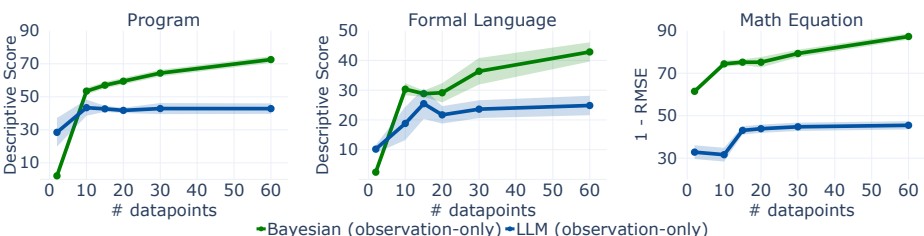

Figure 2: Observation-only results across three black-box types. We compare GPT-4o (blue) to Bayesian inference (green). The horizontal-axis represents the number of provided $(x, y)$ pairs. We report $1 - \text{RMSE}$ for Math Equation and descriptive score for Program and Formal Language.

We first establish the reference performance achievable by the Bayesian model in each setting. These three settings were selected in part because they are all cases where previous work has defined inference algorithms that make it possible to approximate the posterior distribution over hypotheses as more observations becomes available (Rule et al., 2024; Yang & Piantadosi, 2022; Foster et al., 2019). As shown in Figure 2, the Bayesian models (green) consistently improve with the increased number of observations across all three tasks. On the other hand, while the LLM reverse-engineer (blue) starts off with higher performance for Program and Formal Language, potentially leveraging its prior knowledge, it peaks at 10 observations and struggles to use the extra observations thereby causing performance to plateau. We also calculate repeated measures ANOVAs (Girden, 1992) for each black-box type and found significant Model $\times$ number of datapoints interactions for Program ($F(5, 10) = 51.9$, $p < 0.001$), Formal Language ($F(5, 10) = 11.8$, $p = 0.001$), and Math Equation ($F(5, 10) = 8.7$, $p = 0.002$), showing that the Bayesian inference algorithms increasingly outperformed LLMs with additional observations. Details for the ANOVAs are in Appendix D.1.

## 4.2 INTERVENTION IS CRUCIAL FOR THE LLM TO REFINE HYPOTHESES

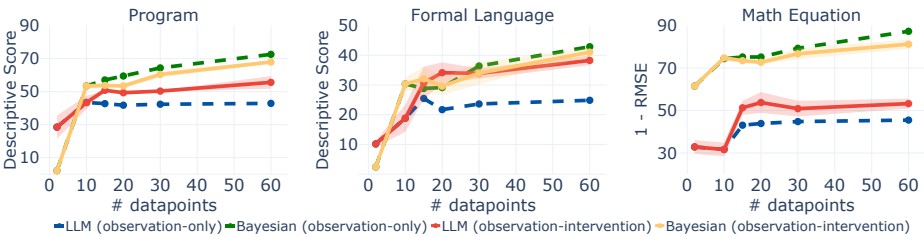

Figure 3: Observation-intervention results across three black-box types. Red: observations and interventions by GPT-4o. Yellow: taking the observation-intervention collected from GPT-4o as observations for the Bayesian inference algorithms. Dashed lines: observation-only reference for GPT-4o (blue) and Bayesian inference (green).

In Figure 3, we compare the performance of models with access to only the observations (dashed lines) against using the data that is actively collected through intervention (solid lines). We observe that enabling the LLM to actively intervene significantly improves performance (red) over observation-only (dashed blue). Through intervention, the LLM consistently improves as more data becomes available across all three black-box types, consistent with prior results on passive learning (Ostrovski et al., 2021). To assess the quality of the interventions, we provide the LLM-collected intervention data to the Bayesian model as observations, akin to the passive yoked data studied in Markant & Gureckis (2010; 2014). Our results indicate that while the interventions are beneficial to the LLM, they are not universally more informative, paralleling findings in human active learning (Markant & Gureckis, 2010; 2014). This gap was statistically significant, as shown by an ANOVA for each black box type: Program ($F(5, 10) = 23.9$, $p < 0.001$), Formal Language ($F(5, 10) = 7.9$, $p = 0.003$), and Math Equation ($F(5, 10) = 14.9$, $p < 0.001$).

## 4.3 IDENTIFYING THE VALUE OF GENERATING THE INTERVENTION DATA

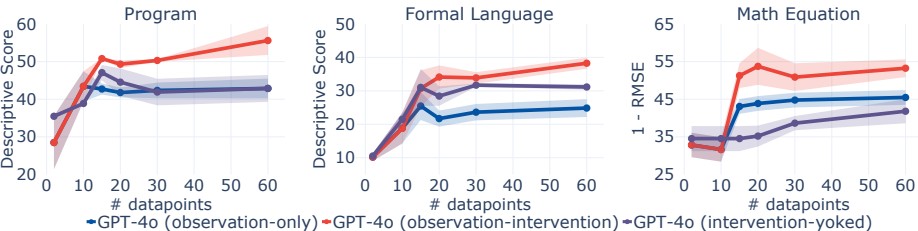

Figure 4: Comparing intervention-yoked results with observation-only and observation-intervention across three black-box types.

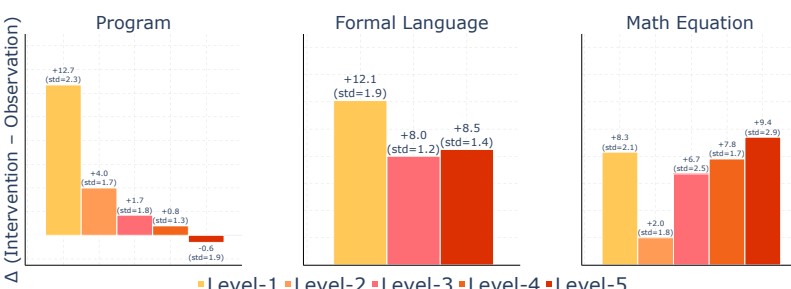

Figure 5: Descriptive scores for five different complexity levels across 3 seeds.

The improvement in performance produced by the interventions could have two sources: it could be that the resulting data are more informative, or that the process of generating interventions itself helps the model. To study this, we adopt the passive-yoked design that Markant & Gureckis (2010; 2014) used to study human learning, where the data generated via active learning by one group of participants is presented to another group of participants as passive observations. In Figure 4, we compare GPT-4o across three conditions: **observation-only** (blue), **observation-intervention** (red), and **intervention-yoked** (purple) where GPT-4o only passively observes the interventional data without the verbalization and analysis that are used to construct such data. Results consistently show that the intervention-yoked setting leads to lower performance compared to the observation-intervention setting across all three black-box types. This shows that active learning is more beneficial than passive-yoked learning in part because it allows the LLM to dynamically refine its hypothesis in response to its own interventions.

## 5 ANALYSIS

### 5.1 ESCAPING THE FAILURE MODES: OVERCOMPLICATION & OVERLOOKING

To understand how intervention improves LLM performance, we analyze common failures by sampling 20 failed examples (scoring below 2 out of 10 points) from the observation-only experiment, which were inspected by human experts. We provide more details in Appendix G.1. We identify two major failure modes: 1) *overcomplication*, where the LLM excessively interprets the data, resulting in unnecessarily complex hypotheses, and 2) *overlooking*, where the LLM inadequately leverages available information, leading to poorly reasoned hypotheses. We classified 20 randomly sampled examples for each black-box into the two failure modes or "Not Applicable" by human annotation. Results show that for Program the failures are predominantly from overcomplication (17 cases out of 20) whereas Math Equation contains more overlooking failures (16 cases out of 20). The failures are more evenly distributed for Formal Language, with 8 examples classified as overcomplication, 11 examples as overlooking, and 1 example as "Not Applicable". We provide examples for these failure mode in Appendix G.2.

Notably, we find that the impact of interventions on alleviating these two failure modes is contingent upon the complexity of the reverse-engineering task itself. For each of the three specific domains we study, we include a brief characterization of complexity in Appendix M. Within each domain, we observe that the complexity of the reverse-engineering problem instance characterized by $f^*$ governs the extent to which interventions rectify overcomplication and overlooking. In Figure 5, we show that performance improvements from intervention on Program diminish as task complexity increases for black-box systems dominated by the *overcomplication* failure mode. In contrast, actively collected data is more beneficial when addressing challenging black-box instances dominated by the *overlooking* failure mode, such as Math Equation. For Formal Language, where both failure modes frequently occur, we observe consistent improvements across all complexity levels.

**Case study.** Figure 6 demonstrates how an LLM progressively updates its hypothesis through active interventions to ultimately reverse-engineer the underlying mechanism of a black-box system using a Formal Language black-box intervention example, where GPT-4o strategically designs subsequent queries to validate its current belief about the system. In contrast, under the observation-only scenario, the model remains trapped in identifying spurious patterns from passively observed data

**Black-Box: Formal Language Rule = A^n B^n C^(2n)**

```
AAAABBBBCCCCCCCC      is generated by the black box
AAABBBCCCCCC          is generated by the black box
{more observations}                                              Observations

Hypothesis: the black box generates strings that follow a pattern where [...]
- If not empty, it consists of a sequence of 'A's followed by 'B's followed by 'C's.
- The number of 'A's and 'B's can vary, but the number of 'C's tends to be larger.
Let's test strings with a different count of 'A's, 'B's, and 'C's to see if it is generated by the black box.
Test: AABBBCCCC      Black-box: cannot be generated
Test: AAABBBCCCC     Black-box: cannot be generated
Test: ABCCCC         Black-box: cannot be generated
Test: AABBCCCC       Black-box: can be generated
Test: AAAABBBBCCCCCCCC Black-box: can be generated            Intervention Round N

Let's review the observations and tests conducted so far to refine our understanding of the black box's rules:
- A's and B's in the generated strings is equal; C's is exactly double the number of A's (or B's)
- AABBBCCCC:   Not generated, indicating the number of A's and B's must be equal. [...]
- ABCCCC:      Not generated, reinforcing the rule that C's must be exactly double the number of A's (or B's).
- AABBCCCC:    Generated, confirming the pattern of equal A's and B's, and C's being double.
Hypothesis: the black box generates strings that follow these rules:
- If not empty, it consists of a sequence of 'A's followed by 'B's followed by 'C's.
- The number of 'A's and 'B's must be equal.
- The number of 'C's must be exactly double the number of 'A's (or 'B's).
Let's test a string that follows this pattern with a different count of A, B, and C.
Test: ABCC           Black-box: can be generated             Intervention Round N+1
```

Figure 6: Case study example. GPT-4o updates the hypothesis using intervention on Formal Language black box. Yellow: GPT-4o states the hypothesis based on the passive observations in round $N$ and updates the hypothesis in round $N + 1$. Red: test case. Teal: black box response.

and lacks a meaningful way to assess its own uncertainty. Through active interventions, the LLM iteratively tests and revises its hypotheses after encountering failures, gradually reducing uncertainty and converging toward an accurate understanding of the black-box mechanism. We add another example to explain the differ between observation-only and observation-intervention in Appendix F.2.

## 5.2 INTERVENTION STRATEGIES

Similar to how LLMs use chain-of-thought reasoning (Wei et al., 2022) to solve complex tasks, we allow the LLM to state its hypotheses and analyze the observations before constructing the query. We investigate how different reasoning strategies impact the effectiveness of intervention. We compare four strategies: 1) Intervention: no reasoning before constructing the query, 2) Descriptive Intervention:

| Black Box | Intervention | Descriptive Intervention | Functional Intervention | Analyze-then-Query Intervention |
|---|---|---|---|---|
| Program | 43.4 | 47.6 | 19.2 | **50.8** |
| Formal Language | 24.1 | 28.6 | 22.8 | **34.7** |
| Math Equation | 34.8 | 38.8 | **39.9** | 38.0 |

Table 1: Comparison of the four intervention strategies. We use Analyze-then-Query as the main intervention strategy.

describing the current hypothesis about the black-box, 3) Functional Intervention: verbalize the black-box implementation as a Python program (Li et al., 2025; Luo et al., 2025), and 4) Analyze-then-Query: allowing the LLM to analyze data and state a hypothesis freely. Throughout our experiments, we allow the LLM to reason once every five queries[1]

As shown in Table 1, allowing the LLM to reason generally improves the effectiveness of intervention regardless of the strategy. However, the most effective intervention typically requires the LLM to carefully analyze past observations and strategically plan subsequent steps to acquire more informative data from the black-box. Interestingly, while structured reasoning in functional intervention (Li et al., 2025; Luo et al., 2025) is known to improve performance in formal reasoning tasks, it does not produce additional improvement in the context of reverse-engineering. This suggests that the LLM reverse-engineering abilities may differ from its formal reasoning capabilities.

## 5.3 TRANSFERRING TO ANOTHER LLM

We also examine whether interventional data actively collected by one LLM (GPT-4o) can effectively transfer and benefit another LLM (Llama-3.3-70B-Instruct). This is relevant to whether AI scientists can transfer their experiments and findings successfully to another AI scientist. The goal of

---

[1]This is a tunable hyperparameter. We fixed it early in the project based on a balance between performance, cost, and runtime.

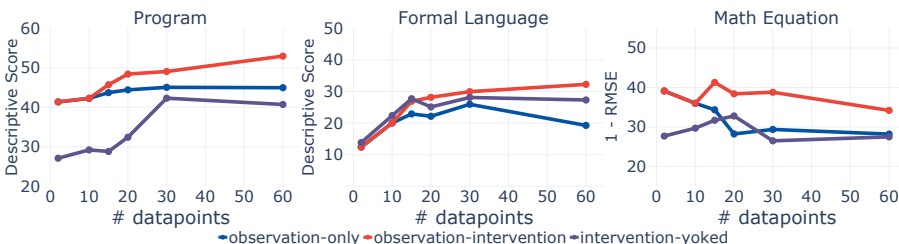

Figure 7: Intervention data transfer results. Red: Llama-3.3-70B-Instruct performing intervention. Blue: Llama-3.3-70B-Instruct using observations only. Purple: using interventional data from GPT-4o as observations for Llama-3.3-70B-Instruct.

this study is to test whether a model that cannot formulate useful queries on its own can nevertheless improve when supplied with high-quality intervention data generated by a stronger model. Adopting a similar passive-yoked design, we compare three scenarios for Llama-3.3-70B-Instruct (Grattafiori et al., 2024): **observation-only**, **observation-intervention**, and **intervention-transfer**, where the interventional data is collected by GPT-4o. As shown in Figure 7, the intervention-transfer scenario achieves performance comparable to or slightly better than the observation-only baseline but consistently underperforms Llama's own intervention (observation-intervention). This suggests that while the intervention data from GPT-4o is informative, the effectiveness diminishes when transferred to a different LLM, showing that the benefit from intervention is model-specific.

## 5.4 REVERSE ENGINEERING ABILITIES ACROSS DIFFERENT CATEGORIES OF LLMS

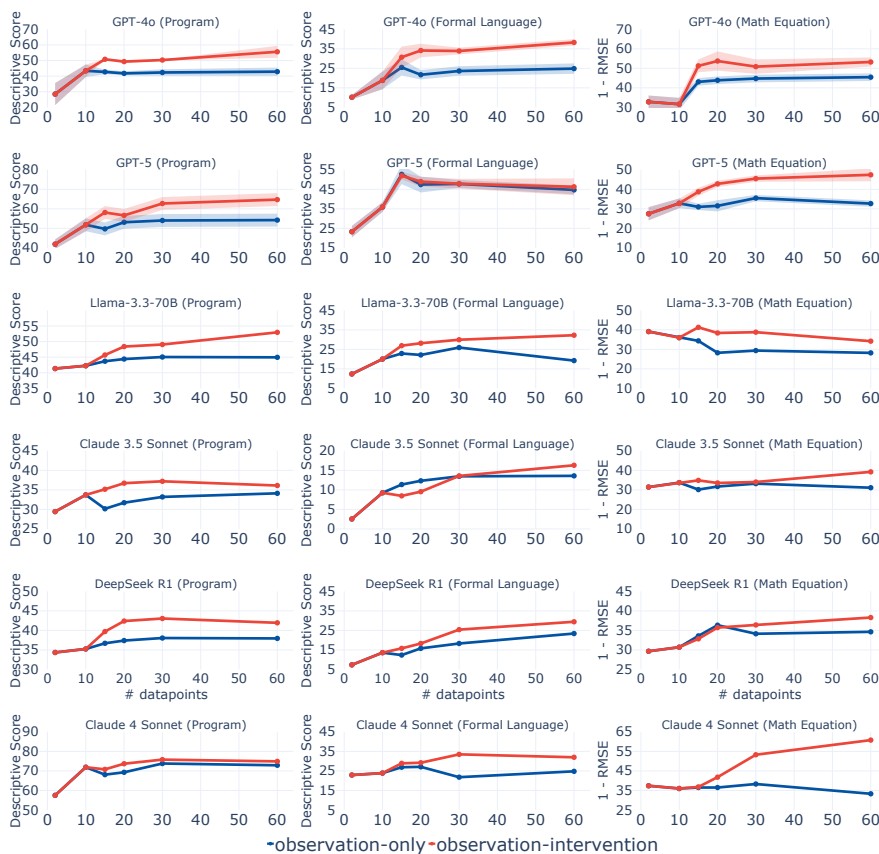

Figure 8: Results of reverse engineering abilities across different categories of LLMs. We report Llama-3.3-70B-Instruct, Claude 3.5 Sonnet, Deepseek R1, GPT-5, and Claude 4 Sonnet. Due to the cost limit, we only run 3 seeds for GPT-4o and GPT-5. We use GPT-4o to judge the results.

In Figure 8, we report observation-only and observation + intervention results across Llama-3.3-70B-Instruct, Claude-3.5-Sonnet, DeepSeek-R1, GPT-5, and Claude-4-Sonnet. Across nearly all black-box types and models, actively refining hypotheses through intervention consistently improves models' understanding of the underlying dynamics. Notably, DeepSeek-R1, Claude-4-Sonnet, and GPT-5—enabled by long-form reasoning—can continue extracting useful information even in passive settings by exploring a wider range of hypotheses. However, despite these advantages, frontier reasoning models do not substantially outperform non-reasoning models (e.g., GPT-4o, Llama-3.3-70B-Instruct, Claude-3.5-Sonnet), except for Claude-4 on Program, underscoring the current limits of reasoning-based approaches for reverse engineering.

## 6 LIMITATIONS AND FUTURE DIRECTIONS

In this paper, we have discussed the inabilities and failure modes of LLMs in reverse-engineering black-boxes. However, the three black-box types we studied represent only a narrow slice of possible tasks, even within controlled settings. A more comprehensive assessment will require expanding and scaling up the evaluation suite to probe LLMs' reverse-engineering abilities across a broader spectrum of scenarios. In addition, we have assumed idealized, noise-free black-boxes and fully trustworthy data—a condition that is rarely met in real scientific practices. An important next step is to relax this assumption and rigorously test LLM robustness in the presence of noise and uncertainty. As our paper discuss extensively on the failure modes of LLMs, we leave open the question: "*How can we train LLMs to become effective reverse engineers?*", which includes enhancing the LLM's ability to perform correct inference from passive observations and to conduct optimal experiments. In particular, what kinds of data and algorithms are needed to train such a model (for example, reinforcement learning using black-box environments), and can improvements in one domain generalize to broader scientific automation tasks? Finally, we have demonstrated that the actively acquired data by one LLM may not be useful for another LLM, pointing to the issue of *experience transferability*. Just as many major scientific advances have relied on effective human collaborations, so too may future automation of scientific discoveries depend on resolving this issue for LLM collaborations. We also note that different LLMs exhibit distinct but inconsistent querying behaviors during interaction, and a more systematic characterization of these patterns is an important direction for future work. Understanding and quantifying the impact of this limited transferability of knowledge may be crucial as multi-agent systems become prevalent, and it will be essential to design such systems with effective communication baked in.

## 7 CONCLUSION

In this paper, we identified and formalized the reverse-engineering problem as a core ability and prerequisite for performing a reliable scientific discovery. We showed that current LLMs still struggle to effectively leverage passive observations even on seemingly simple and controlled black-boxes. Allowing LLMs to actively collect intervention data improves performance, but still falls short of closing the gap with Bayesian inference, casting doubt on the promise of truly reliable AI scientists. Through extensive analysis, we identified issues such as overcomplication and overlooking and illustrate how intervention can mitigate such failures. Despite the effectiveness of intervention, our analysis revealed that the intervention data collected by LLMs were primarily beneficial to the models themselves, rather than being objectively informative or transferable to other models.

Altogether, our paper directly assesses the ability of LLMs to infer underlying causal structures and mechanisms through controlled reverse-engineering experiments. This capacity mirrors the fundamental scientific discovery process, which relies heavily on identifying hidden relationships and principles behind observed phenomena. Consequently, if an LLM cannot reliably reverse-engineer even simple or controlled systems, this raises concerns regarding its dependability in addressing more complex and ambiguous scientific challenges. Evaluating an LLM's reverse-engineering ability provides a concrete and principled way to assess its capacity for scientific reasoning, helping us understand whether such models possess the foundational skills required to function as dependable AI scientists.

## 8  ETHICS STATEMENT

This work investigates how language models can reverse-engineer black-box systems in fully synthetic domains such as programs, formal languages, and mathematical equations. Our study does not involve human subjects, sensitive or proprietary data, or any real-world systems. While reverse-engineering methods in general could raise concerns if applied to sensitive settings, our research design deliberately avoids such cases by restricting all experiments to controlled, non-sensitive environments. We do not identify direct ethical issues beyond the standard considerations for computational research.

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

## A APPENDIX

## B BLACK BOX DESIGNS

**Program**  We used 100 list-mapping program instances from (Rule et al., 2024) to design the Program black-box API. Each black-box instance represents as a symbolic program defined in a domain-specific language (DSL). We implemented an interpreter pipeline that parses DSL expressions into abstract syntax trees and compiles them into executable Python code.

Each black-box supports two modes: `observation` (observation-only) and `intervention` (observation-intervention). In the `observation` mode, the black-box takes a random input list and returns the output produced by the underlying symbolic program, generating paired observational data:

$$\text{input list} \rightarrow \text{program execution} \rightarrow \text{output list}$$

In the `intervention` mode, the LLM queries an input or explicitly specifies an input-output pair. The black-box generates the output list or evaluates whether the given output matches the internally computed output and provides clear feedback:

$$\text{Feedback} = \begin{cases} \text{"output} \Rightarrow \text{Correct"}, & \text{if the provided output matches the program output,} \\ \text{"output} \Rightarrow \text{Incorrect"}, & \text{otherwise.} \end{cases}$$

**Formal Language**  We followed (Yang & Piantadosi, 2022; McCoy & Griffiths, 2023) to implement a collection of 46 formal language instances to construct our formal language black-box, each instance being capable of generating strings according to specific symbolic rules (*e.g.* $A^n B^n$). Each black-box instance behaves as an API from a generative model, operating in two modes: `observation` and `intervention`.

In the `observation` mode (observation-only), the black-box randomly produces valid strings from its underlying rule, explicitly labeling each as generated output, for example:

$$\text{"AAAABBBB" is generated by the black-box.}$$

In the `intervention` mode (observation-intervention), the LLM submits a specific string query to the black-box, which evaluates whether the string complies with its rule. The black-box responds clearly, indicating either acceptance or rejection:

$$\text{Response} = \begin{cases} \text{"[string] is generated by the black-box"}, & \text{if the strings compile with the rule,} \\ \text{"[string] cannot be generated by the black-box"}, & \text{otherwise.} \end{cases}$$

To avoid generating infinite strings, we imposed a maximum character length of 64 for all single characters generated by the black-box.

**Math Equation**  For the math equation, we implemented the CES utility model as the black-box, designing it as a generative model capable of generating observational data or responding to queries from an LLM. The utility function of CES is mathematically defined as:

$$U = \left( \sum_i a_i x_i^r \right)^{\frac{1}{r}},$$

where the weights $a_i$ satisfy the constraint $\sum_i a_i = 1$, the parameter $r$ controls the substitution elasticity, and $x_i$ represents the quantities of goods in a basket.

CES black-box also provides two operational modes: `observation` (observation-only) and `intervention` (observation-intervention). In the `observation` mode, the black-box randomly samples two baskets (each a list of good quantities) and computes their utilities using the CES formulation. It then returns the preference outcome indicating which basket is preferred based on

higher utility:

$$\text{Preference} = \begin{cases} \text{Basket1,} & U(\text{Basket1}) > U(\text{Basket2}), \\ \text{Basket2,} & U(\text{Basket1}) < U(\text{Basket2}), \\ \text{equal utility,} & U(\text{Basket1}) = U(\text{Basket2}). \end{cases}$$

In the `intervention` mode, an external model explicitly queries the black-box by specifying two baskets. In addition, the external model can also provide an estimated preference. The CES black-box internally evaluates the utilities based on the specified baskets and returns the actual preference outcome or feedback indicating whether the provided estimate was "correct" or "incorrect".

## C   BAYESIAN MODELS AS THE 'OPTIMAL' REFERENCE

We employ Bayesian models as an oracle for optimal reverse-engineering against which we may assess the capabilities of LLMs. Unlike LLMs, Bayesian models explicitly perform probabilistic inference within a clearly defined hypothesis space, systematically updating posterior beliefs using the Bayes rule to identify the underlying mechanism that best explain observed data. Under the critical assumption that the true underlying rule resides within this hypothesis space (that is, the standard assumption of a well-specified prior), Bayesian models serve as an optimal reference standard in our experimental setting. We hypothesize that LLMs, when provided only with passive observational data, are unable to effectively utilize available information due to their inherent reliance on prior knowledge, resulting in significantly lower performance compared to the Bayesian optimal standard. However, allowing LLMs to actively intervene and collect data can substantially reduce the performance gap. For each of the three black-box systems evaluated, we replicated the Bayesian models from their original studies, adapting them to closely match our experimental conditions. Specifically, we provide Bayesian models with observed data generated by our black-box systems as an ideal reference. We also provide Bayesian models with the actively collected data from LLMs intervention to assess the informativeness of the data gathered by LLMs. To ensure rigorous comparability, we applied identical evaluation methodologies to both the Bayesian models and LLMs.

**Program**   We used the Bayesian inference approach from Rule et al. (2024) to establish an optimal reference for list-mapping program black-box. Specifically, we utilized their MetaProgram Learner, which performs Bayesian inference over symbolic metaprograms that generate target programs from observed data.

Given observational data $D$, consisting of input-output pairs generated by symbolic programs, the MPL computes the posterior distribution over candidate hypotheses (metaprograms) $H$ according to the Bayes rule:

$$P(H \mid D) \propto P(D \mid H) \cdot P(H).$$

The prior distribution $P(H)$ integrates two complementary sources of simplicity bias: the meta-program prior $P_{\mathcal{M}}(H)$ and the induced program prior $P_{\mathcal{P}}(\widetilde{H})$. This combined prior is defined as:

$$P(H) \propto \exp\left( \frac{\ln P_{\mathcal{M}}(H) + \ln P_{\mathcal{P}}(\widetilde{H})}{2} \right),$$

where $\widetilde{H}$ denotes the program compiled from the metaprogram $H$.

The likelihood $P(D \mid H)$ measures the consistency of a meta-program $H$ with the observational data provided, incorporating a noise model to accommodate minor discrepancies between the model predictions and observations.

**Formal Language**   We adopted the Bayesian inference approach from (Yang & Piantadosi, 2022) as an optimal reference model to determine the theoretical upper bound on the learnability of formal language rules from the observations generated by our black-boxes or from the interventions queried by LLM. Specifically, we provided strings generated by our formal language black-boxes

as observational data to the Bayesian model, which then inferred the underlying symbolic grammar rules.

Just as before, the Bayesian inference framework defines the posterior distribution over candidate hypotheses conditioned on observed data using Bayes' rule:

$$P(H \mid D) \propto P(D \mid H) \, P(H),$$

where $H$ represents a candidate hypothesis (grammar or probabilistic program), $D$ represents the observed string data generated by the black-box, $P(H)$ represents the prior probability reflecting initial beliefs about the simplicity and plausibility of hypotheses, and $P(D \mid H)$ denotes the likelihood of observing data $D$ given hypothesis $H$.

The Bayesian model uses a structured prior $P(H)$, assigning higher probabilities to simpler, more concise grammars or symbolic programs. As observational data increases, Bayesian updating systematically refines prior beliefs into posterior distributions, enhancing the probability assigned to grammars that best explain the data. Formally, each new observed string updates the posterior, shifting probability mass toward hypotheses consistent with the cumulative dataset. By leveraging this Bayesian inference mechanism, we quantify the upper bound of the learnability of the observations, thus providing a rigorous baseline to evaluate LLM's effectiveness in utilizing the same observational data.

**Math Equation**  To infer the parameters of the CES utility model from the observations provided, we followed (Foster et al., 2019) by employing a Bayesian inference approach explicitly conditioned on these observations. Bayesian inference integrates observed data with prior beliefs, updating these beliefs into posterior distributions to progressively improve parameter estimates. Initially, we specified prior distributions for the model parameters:

$$\rho \sim \mathrm{Beta}(\rho_0, \rho_1),$$
$$\alpha \sim \mathrm{Dirichlet}(\alpha_{\mathrm{conc}}),$$
$$\mathrm{slope} \sim \mathrm{LogNormal}(\mathrm{slope}_\mu, \mathrm{slope}_\sigma).$$

Given pairs of consumption bundles $(d_1, d_2)$ and the corresponding observed user preferences $y$, the Bayesian framework models these preferences probabilistically through a censored sigmoid-normal likelihood:

$$y \sim \mathrm{CensoredSigmoidNormal}\left(\mathrm{slope} \cdot (U(d_1) - U(d_2)), \; \mathrm{slope} \cdot \mathrm{obs\_sd} \cdot (1 + \|d_1 - d_2\|_2)\right),$$

where $U(d_1) - U(d_2)$ denotes the utility difference between the two bundles. Here, "censored" refers to applying a sigmoid function to latent utility values and then truncating the results to the observed preference interval (e.g., $[0, 1]$), ensuring that responses remain within these limits.

The posterior distributions are updated via Bayes' theorem by explicitly integrating observational data:

$$p(\rho, \alpha, \mathrm{slope} \mid y, d) \propto p(y \mid \rho, \alpha, \mathrm{slope}, d) \, p(\rho, \alpha, \mathrm{slope}),$$

where $p(\rho, \alpha, \mathrm{slope})$ represents prior distributions and $p(y \mid \rho, \alpha, \mathrm{slope}, d)$ represents the likelihood function given the observations.

While some sources prefer uppercase probability notation such as $P(H \mid D)$, this paper adopts lowercase notation ($p$) consistently for both probability densities and random variables throughout.

Parameter estimation was performed via variational inference (Blei et al., 2017), iteratively optimizing the evidence lower bound (ELBO), defined as:

$$\mathrm{ELBO}(\phi) = \mathbb{E}_{q_\phi}\left[\log p(y \mid \rho, \alpha, \mathrm{slope}, d)\right] - D_{\mathrm{KL}}\left(q_\phi(\rho, \alpha, \mathrm{slope}) \parallel p(\rho, \alpha, \mathrm{slope})\right),$$

where $q_\phi$ denotes the variational posterior distribution used to approximate the true posterior distribution.

Thus, as additional observational data are incorporated, Bayesian inference continually updates prior beliefs into posterior distributions, systematically refining parameter estimates toward their true underlying values.

## D  STATISTICAL SIGNIFICANT TESTS

### D.1  REPEATED-MEASURES ANOVA

To statistically evaluate the interaction between models (Bayesian vs. LLM) and steps, we calculated the repeated-measures ANOVAs. Each black-box instance involved multiple repeated measurements corresponding to different steps. Letting $Y_{ijk}$ represent the performance score for subject $i$, models $j$ (Bayesian or LLM), and step $k$, the repeated-measures ANOVA model can be expressed as:

$$Y_{ijk} = \mu + S_i + M_j + T_k + (M \times T)_{jk} + \epsilon_{ijk}$$

where $\mu$ is the mean in all measurements, $S_i$ represents the random effect of the subjects (individual seeds), $M_j$ denotes the main effect of the model, $T_k$ is the main effect of steps, $(M \times T)_{jk}$ is the interaction between the model and the step, and $\epsilon_{ijk}$ represents residual error.

The ANOVA decomposes the total variance into these distinct sources. Specifically, the significance of the interaction of the Step Method $\times$ was determined by calculating the corresponding F-statistic:

$$F = \frac{MS_{(M \times T)}}{MS_{error}}$$

where $MS_{(M \times T)}$ is the mean square for the Method $\times$ Step interaction, and $MS_{error}$ is the residual mean square. Significance was assessed using an $F$-distribution with numerator degrees of freedom equal to $(J - 1)(K - 1)$, where $J$ is the number of method levels and $K$ is the number of steps, and denominator degrees of freedom equal to $(I - 1)(K - 1)$, where $I$ is the number of subjects.

## E  PROMPTS

### E.1  INTERVENTION PROMPT

```
In this task, you are given a ``black box'' and need to determine
its inner workings.
{black box information}
You will have a series of turns to interact with the black box. On
each turn, you can either gather more information or test your
hypothesis. To gather more information, {query instruction}, and
obtain a result.
To test your hypothesis, {test instruction}. All the information
gathered across all the turns is used to reverse engineer the
black box. Throughout the process, you can decide whether the
gathered information is sufficient to correctly identify the
workings of the black box, in which case you can stop. Otherwise,
you need to continue the interaction. Concretely, you can perform
one of the following actions at each turn: 1) query, 2) test, or
3) stop.

Provide a *thorough reasoning* before performing the action.
Leverage the past observations to design your next query and make
your hypothesis as accurate as possible. Below is the format for
each action.

Query:
```query
List[int]
```

Test:
```

```test
List[int]
List[int]
```

Stop:
```stop
```

Note that you should only perform one of the actions above with one input example in your response.

Below are your past observations of the black box:
{observations}
Response:

## E.2 EVALUATION PROMPTS

**Program (judge):**

In this task you will be given a ground truth program and pseudocode that you need to evaluate. You will output a score for the quality of the pseudocode based on a set of assessment criteria.

Below is the ground truth program:
{ground_truth}

Evaluate the quality of the following pseudocode:
{response}

Score the above pseudocode against the ground truth program based on the following criteria (total 10 points):
1. Does the provided pseudocode correctly specify the implementation of the ground truth program and manipulate the variables in the same way? Ignore the programming language difference. [5 point]
2. Does the provided pseudocode specify the implementation in the most simple and straightforward way without extra unused parts (Occam's Razor principle) [5 point]

Explain your judgement and return the final score with the type float and following the format below:
```judgement
YOUR JUDGEMENT HERE
```
```score
YOUR SCORE HERE
```

Response:

**Formal Language (judge):**

In this task, you will be given a ground truth formal language and a proposed rule describing that formal language, which you need to evaluate for quality. You will then output a score based on a set of assessment criteria.

Below is the ground truth formal language:

```
{ground_truth}

Evaluate the quality of the following formal language rule:
{response}
Score the above formal language rule against the ground truth
formal language based on the following criteria (total: 10
points):
1. Does the provided rule correctly generate the examples given in
the ground truth? Your score is determined by how many examples
are correctly generated out of the total number of examples. [3
points]
2. Does the provided rule correctly reverse engineer the ground
truth formal language? [5 point]
3. Is the provided rule in the most simple and straightforward way
without extra unused parts (Occam's Razor principle)? Note: If the
provided rule is completely incorrect, you should give 0 point for
this criterion. [2 point]

Explain your judgement and return the final score with the type
float and following the format below:
```judgement
YOUR JUDGEMENT HERE
```
```score
YOUR SCORE HERE
```

Response:
```

**Math Equation (judge):**

```
In this task, you are provided with a ground truth CES utility
function and a CES utility function predicted by a model.

Your task is to evaluate the quality of the predicted utility
function based on a set of assessment criteria and output a score.

The ground truth utility takes this form:
U(\\mathbf{{x}}) = \\left(\\sum_{{i=1}}^n a_i \\cdot
x_i^{{\\text{{rho}}}}\\right)^{{1/\\text{{rho}}}}

The utility depends on the following parameters:
1. a_i: float rounded to the first decimal point and should sum up
to 1. (Note that there will be multiple a_i's.)
2. rho: float rounded to the first decimal point.

Below is the information about the ground truth utility function:
{ground_truth}

Evaluate the quality of the following predicted the parameters of
the utility function:
{response}

Score the predicted utility function against the ground truth
using the following criteria (total 10 points):
1. Is the predicted utility function has a correct rho? [2 points]
2. Compare the predicted utility function to the ground truth, how
many a_i's are correct (order matters)? This will give us an
accuracy percentage. The score for this bullet should be the
accuracy percentage times the total allocated 6 points [6 points]
```

3. In the predicted utility function, do the unknown parameters a_i sum up to 1 and do the number of a_i's match the number of goods? [1 point]
4. Does the predicted utility function express the function in a simple and straightforward way without any unnecessary elements (adhering to the Occam's Razor principle)? [1 point]

Explain your judgement and return the final score with the type float and following the format below:
```judgement
YOUR JUDGEMENT HERE
```

```score
YOUR SCORE HERE
```

Response:

**Descriptive Evaluation:**

In this task, you are given a ``black box`` and need to determine its inner workings.
{black box information}

Below are some past observations from the black box:
{observations}

Your task is to reverse engineer the rule underlying {more detailed instructions} in the following format:
```Rule
YOUR RULE HERE
```

Response:

**Function Implicit Evaluation:**

In this task, you are given a ``black box`` and need to determine its inner workings.
{black box information}
Below are some past observations from the black box:
{observations}

{More detailed instructions}

Output your generated string in the following format:
```output
YOUR RESPONSE HERE
```

Response:

# F ADDITIONAL EXAMPLES

## F.1 DESCRIPTIVE EVALUATION

```
Language rule:  "AAAA"
Language examples:  "A, AA, AAA, AAAA"
Score of 30:
LM's answer:  All strings consist only of the character 'A', and
the lengths follow the pattern of powers of 2.

Score of 50:
LM's answer:  The formal language generates all non-empty strings
consisting only of the character 'A'. In other words, it generates
A^n where n ≥ 1.
```

## F.2 HYPOTHESIS REFINEMENT EXAMPLE

In the example below, the model forms a hypothesis about the pattern and then queries the black box to test the hypothesis to mitigate its own uncertainty.

```
Language:A^(2^n)

Interventions:
Based on the observations provided, we can see that the black
box generates strings consisting solely of the character 'A'. The
strings that have been generated by the black box vary in length,
but they all share the common characteristic of being composed
entirely of the letter 'A'.
...
From these observations, it seems that the black box generates
strings of 'A's of any length.  To further test this hypothesis,
I will query the black box with a string of 'A's of a length that
has not been observed yet:  AAA

Black box's response: AAA cannot be generated by black box.
```

# G COMMON FAILURE MODES

## G.1 HUMAN ANNOTATION

To systematically analyze LLM's failure modes, we defined an LLM reverse-engineering attempt as a failure if its descriptive score was below 2 out of 10, according to our descriptive evaluation rubric. For each black-box type, we randomly selected 20 representative failure cases from the observation-only setting. We have two human experts independently reviewed these examples, categorizing each case based on the nature of the error. Any disagreements were resolved through discussion. Finally, human annotators identified two common failure modes: overcomplication and overlooking.

## G.2 OVERCOMPLICATION & OVERLOOKING EXAMPLES

Across the three black-box types, we find that overcomplication is a common failure mode, particularly in the Program, while overlooking most often occurs in Math Equation. For Formal Language, both overcomplication and overlooking are observed when LLMs fail at reverse engineering. In Program (see Table 2), we observe that the model introduces an extra operation that never appears in any input–output pair. In the Language overcomplication example (see Table 3), the model adds restrictive structural conditions, such as specific block lengths or mandatory patterns that do not appear in

the observation set. We also observe overlooking failures in Language (see Table 4), where the LM outputs a rule that contradicts basic strings in the data due to not fully using the observed examples. For Math Equation, we usually observe overlooking failures. For example, in Table 5, the model uses only a subset of the basket comparisons and ignores combinations that directly determine the CES parameters.

We also compare the failure examples for different models. In the overcomplication example from GPT-5 (Table 6), the model asserts that B-blocks must have lengths that are multiples of three, a restriction that does not appear in any observed string. In the corresponding example from Claude-4 (Table 7), the model imposes broad conditions such as forbidding the substring AAA or requiring the presence of A, even though none of these requirements are supported by the observations. For overlooking, the GPT-5 example in Table 8 fails because the model uses only a subset of basket comparisons and ignores pairs that uniquely identify the CES parameters. In the Claude-4 example (Table 9), the model outputs a CES function with arbitrary parameter choices rather than checking whether the function is consistent with the provided observations.

## H  COMPLEXITY CATEGORIZATION

We rank the complexity level from $1-5$. Each black-box type includes multiple instances of varying task complexity.

**Program.** The complexity level is determined based on the number of operations, which ranges from $1-12$. Instances with fewer than 2 operations are classified as complexity level 1 ($complexity-1$), those with fewer than 4 operations as $complexity-2$, fewer than 6 operations as $complexity-3$, and fewer than 8 operations as $complexity-4$. Due to the limited number of remaining examples, all others are grouped into the highest complexity level ($complexity-5$).

**Formal Language.** Instead of using five complexity levels, we divided the Formal Language instances into three levels, drawing on insights from (La Torre et al., 2007). Specifically, we categorized regular language instances as complexity-1 black-boxes, context-free languages as complexity-3, and context-sensitive languages as complexity-5.

**Math Equation.** We categorize complexity levels according to the number of goods involved, ranging from 2 to 6. Specifically, instances with 2 goods are labeled as $complexity-1$, 3 goods as $complexity-2$, and so on, with instances involving 6 goods classified as the highest complexity level, $complexity-5$.

## I  RELIABILITY AND ACCURACY OF USING GPT-4O AS A JUDGE

The use of LLM-as-Judge has been a common practice to evaluate model generation and GPT-4 level models have been shown to match or exceed human annotation in quality (Liu et al., 2023; Li et al., 2024) for evaluating generated text. In our experiment settings, the LLM judge takes a set of rubrics that sum to a total of 10 points, and the description of the black-box instance to score the model response description of the black-box instance to score the model response. Our implementation further removes the potential vagueness by adding rubrics to evaluate the correctness in a fine-grained manner. The description of the black-box instances are also non-ambiguous to the model as we provide the context in which they need to be interpreted. We show GPT-4o's reliability as a judge by computing Cohen's kappa between GPT-4o and (i) thinking LLMs (OpenAI o3 and Claude-4-Sonnet) and (ii) human annotations. We randomly sample 30 examples (10 for each black-box type) and collect annotations to calculate the Weighted Cohen Kappa score (for ordinal rating). We obtain an overall Weighted Cohen Kappa score of 0.773 for Human, 0.752 for Claude 4, and 0.734 for o3. All the results indicate substantial agreement (Landis JRKoch, 1977) and show the reliability and accuracy of using GPT-4o as a judge.

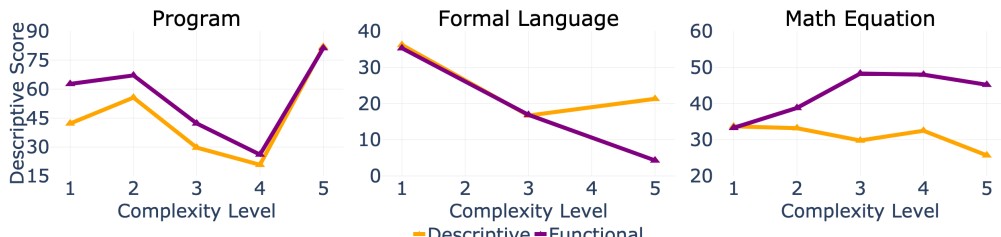

Figure 9: Comparison of descriptive evaluation (yellow) and functional evaluation (purple) across black-box complexity levels.

## J  EVALUATION OF THE REVERSE-ENGINEERING ABILITY

Unlike typical tasks used to benchmark LLMs, such as solving math problems or question answering which are commonly evaluated using accuracies, the reverse-engineering ability is less straightforward. One can assess how well the black-box $f^*$ is recovered by an LLM by: 1) *descriptive* evaluation where the LLM verbalizes the hypothesis to compare to the ground truth and 2) *functional* evaluation which captures how well the LLM emulates the black-box's input-output dynamics and generalizes to unseen examples (Kang et al., 2024). In functional evaluation, the LLM directly predicts the response conditioned on the test query and the past observations and compute accuracy $\texttt{Acc} = \frac{1}{M} \sum_{i=1}^{M} \mathbb{1}[y_i^{\text{test}} = \mathcal{M}(x_i^{\text{test}}, \mathcal{O})]$, without generating the black-box implementation, akin to in-context learning (Brown et al., 2020). As shown in Figure 9, descriptive and functional evaluation trends align for Program across complexity levels. However, we also observe discrepancies of trends between the two evaluations for Formal Language (complexity level 3 to 5) and Math Equation (complexity level 1 to 3), demonstrating that the evaluation protocol and the *format* of the model output may capture different strengths and weaknesses of the model. For Program, we used the original samples from the black box of the list mapping program as test cases (Rule et al., 2024) and ensured that none of these input–output pairs were included in the observations. For Formal Language and Math Equation, we use our deterministic black-box randomly sample 20 test cases per black-box instance.

## K  ANOTHER BLACK-BOX TYPE: BOARD GAME

### K.1  BLACK-BOX DESIGN

We design a connect-$n$ board game ($2 \times 2$ to $9 \times 9$) variant following (Zhang et al., 2024). The black-box is defined by the rules that dictate the winning condition of the game (e.g., "Win by connecting 3 stones in a column."). The LLM can query with a board state and an action, and the black-box responds with the new board state and a game status (win/lose/draw/ongoing). In our black-box design, every game instance exposes two modes—`observation` (observation-only) and `intervention` (observation-intervention)—and uses the symbols X and O to mark the moves of the two players.

**Game definition.**  For a given instance, let the board be a $r \times c$ grid and let $\langle r_{\text{win}}, c_{\text{win}}, d_{\text{win}} \rangle$ denote the required number of consecutive marks needed to win horizontally, vertically, and diagonally, respectively. During play the black-box tracks the current board state $B$, the active player $p \in \{X, O\}$, and whether the game has ended.

In `observation` mode, an external LLM supplies an *initial* board (or leaves it empty). The black-box generates the following as the outputs:

- the round number,
- the updated board,
- whose move it was last,
- the current game status (*ongoing*, *draw*, *PlayerX_wins*, etc.).

If the move ends the game, the record also names the winner.

In `intervention`, the LLM needs to specify (i) additional pieces to place on the board, (ii) the candidate action it wishes the black-box to take, and (iii) optionally, a predicted follow-up board. The black-box returns the same structured record as in observation mode. If the LLM also proposed a prediction of the next state, the black-box confirms it (*"Correct"*) or explains why it is invalid. For Board Game, we do not have a Bayesian model as the optimal reference for the comparison.

## K.2 BOARD GAME RESULTS

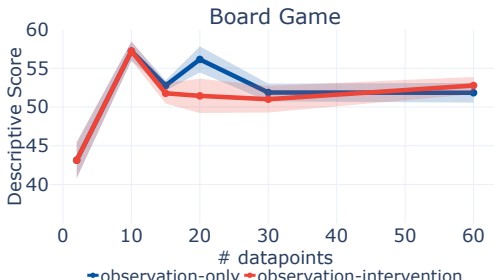

Figure 10: Observation-only and observation-intervention results for Board Game.

In Figure 10, we do not observe the same trends seen in Programs, Formal Language, and Math Equation black-box types. For Board Game, actively collected data does not improve the reverse-engineering performance of the model, indicating that the data gathered is not even significantly informative for the LLM itself. We hypothesize that this is because, to query the black-box, the LLM must (1) generate a board state, (2) propose a next move, and (3) predict the resulting board state, requiring a multi-step reasoning process. These compounded requirements make it challenging for the LLM to probe edge cases or effectively reduce uncertainty about the black-box. This result further highlights a key limitation of current LLMs: When the information signal from the black-box is sparse, actively collected data remain of limited utility.

## L FUNCTIONAL EVALUATION DETAILS

For Program, we used the original samples from the black box of the list mapping program as test cases (Rule et al., 2024) and ensured that none of these input–output pairs were included in the observations. For Formal Language and Math Equation, we use our deterministic black-box randomly sample 20 test cases per black-box instance.

## M COMPLEXITY CATEGORIZATION

We rank the complexity level from $1-5$. Each black-box type includes multiple instances of varying task complexity.

**Program.** The complexity level is determined based on the number of operations, which ranges from $1-12$. Instances with fewer than 2 operations are classified as complexity level 1 ($complexity-1$), those with fewer than 4 operations as $complexity-2$, fewer than 6 operations as $complexity-3$, and fewer than 8 operations as $complexity-4$. Due to the limited number of remaining examples, all others are grouped into the highest complexity level ($complexity-5$).

**Formal Language.** Instead of using five complexity levels, we divided the Formal Language instances into three levels, drawing on insights from (La Torre et al., 2007). Specifically, we categorized regular language instances as complexity-1 black-boxes, context-free languages as complexity-3, and context-sensitive languages as complexity-5.

**Math Equation.**    We categorize complexity levels according to the number of goods involved, ranging from 2 to 6. Specifically, instances with 2 goods are labeled as $complexity - 1$, 3 goods as $complexity - 2$, and so on, with instances involving 6 goods classified as the highest complexity level, $complexity - 5$.

| | |
|---|---|
| **Black-box instance: (lambda (singleton (third $0)))** | |

| | |
|---|---|
| **Observations:** | Input: `[97, 53, 5, 33, 65, 62, 51]`; Output: `[5]` |
| | Input: `[61, 45, 74, 27, 64]`; Output: `[74]` |
| | Input: `[36, 17, 96]`; Output: `[96]` |
| | Input: `[79, 32]`; Output: invalid input |
| | Input: `[90, 77, 18, 39, 12, 93, 9, 87, 42]`; Output: `[18]` |
| | Input: `[71, 12, 45, 55, 40, 78, 81, 26]`; Output: `[45]` |
| | Input: `[61, 56, 66, 33, 7, 70, 1, 11, 92]`; Output: `[66]` |
| | Input: `[90, 100, 85, 80, 0, 78, 63]`; Output: `[85]` |
| | Input: `[31, 93, 41, 90, 8, 24]`; Output: `[41]` |
| | Input: `[28, 30, 18, 69, 57, 11, 10, 40, 65, 62]`; Output: `[18]` |
| | Input: `[38, 70]`; Output: invalid input |
| | Input: `[90, 15, 70, 42, 69]`; Output: `[70]` |
| | Input: `[77, 70, 75, 36]`; Output: `[75]` |
| | Input: `[11, 76, 49, 40, 73, 30, 37, 23]`; Output: `[49]` |
| | Input: `[23, 4, 78, 84]`; Output: `[78]` |
| | . . . |
| | {More observations} |
| | . . . |
| | Input: `[60, 8, 11, 86, 96]`; Output: `[11]` |
| | Input: `[19, 4, 10]`; Output: `[10]` |
| | Input: `[87, 50, 90, 67, 35, 66, 30, 27, 86]`; Output: `[90]` |
| | Input: `[53, 74, 35, 57, 63, 84, 82, 89, 45, 10]`; Output: `[35]` |
| | Input: `[78, 14, 62, 75, 80, 42]`; Output: `[62]` |
| | Input: `[31, 2, 93, 34]`; Output: `[93]` |
| | Input: `[90, 28]`; Output: invalid input |
| | Input: `[21, 42, 54, 7, 12, 100]`; Output: `[54]` |
| | Input: `[89, 28, 5]`; Output: `[5]` |
| | Input: `[81, 68, 77, 87, 9, 3, 15, 81, 24, 77]`; Output: `[77]` |
| | Input: `[15, 50, 11, 47, 14, 4, 77, 2, 24, 23]`; Output: `[11]` |
| | Input: `[61, 26]`; Output: invalid input |
| | Input: `[86]`; Output: invalid input |
| | Input: `[79, 12, 33, 8, 28, 9, 82]`; Output: `[33]` |
| | Input: `[44, 55, 23, 7, 64]`; Output: `[23]` |

| | |
|---|---|
| **Model response: (Overcomplication)** | FUNCTION black_box(input_list): IF length of input_list ¡ 3: RETURN "invalid input" ELSE: **SORT input_list in ascending order** RETURN the third element in the sorted input_list |
| | Explanation: The black box function takes a list of integers as input. If the list contains fewer than 3 elements, it returns "invalid input". Otherwise, **it sorts the list in ascending order and returns the third smallest element** (which is the third element in the sorted list). |

Table 2: Overcomplication example of Program. Red: LLM comes up with a more complex rule of black-box instance.

| **Black-box instance: AnBmAnBm** ($A^n B^m A^n B^m$) | |
|---|---|
| **Observations:** | AAAAAAAAAAAAAA is generated by the black box |
| | AABBBBBBAABBBBBB is generated by the black box |
| | AAAAAAAAABBAAAAAAAAABB is generated by the black box |
| | AAAAAAAABBBBBAAAAAAAABBBBB is generated by the black box |
| | AAAAABBBBBBBAAAAABBBBBBBB is generated by the black box |
| | AAAAAAAABBBBBBBBBBAAAAAAAABBBBBBBBBB is generated by the black box |
| | AAABBBBBBBBBBAAABBBBBBBBBB is generated by the black box |
| | AAAAAAAABBBBBBAAAAAAAABBBBBB is generated by the black box |
| | AAABBBBBBBBBBAAABBBBBBBBBB is generated by the black box |
| | ... |
| | {More observations} |
| | ... |
| | AAABBBBBAAABBBBB is generated by the black box |
| | AAAABBAAAABB is generated by the black box |
| | AAAAAAAAABAAAAAAAAAB is generated by the black box |
| | AAAAAABBBBBBAAAAABBBBBB is generated by the black box |
| | AAAAAABBBBBAAAAAABBBBB is generated by the black box |
| | AAAAAAAAAAAAAAAAA is generated by the black box |
| | AAAAAAAAAABBBBBBBBAAAAAAAAAABBBBBBBB is generated by the black box |
| | AAAAAAAAAABBAAAAAAAAAABB is generated by the black box |
| | AAAAABBBBBBBBBAAAAABBBBBBBBB is generated by the black box |
| | AAAAABBBBBBBBAAAAABBBBBBBB is generated by the black box |
| | AAAABBBBAAAABBBB is generated by the black box |
| **Model response:** **(Overcomplication)** | The formal language generates strings composed of two segments: a sequence of 'A's followed by a sequence of 'B's. **The number of 'A's in the first segment must be equal to the number of 'B's in the second segment,** or the number of 'A's can be zero, resulting in a string of only 'B's. Additionally, the language allows for strings composed entirely of 'A's or entirely of 'B's. The sequences of 'A's and 'B's can be repeated multiple times, as long as the rule of equal counts between 'A's and 'B's in each segment is maintained. |

Table 3: Overcomplication example for Formal Language. Red: The LLM predicts a rule that is more complex than necessary for this black-box instance. In this example, it incorrectly reverse the number of As to be equal to the number of Bs.

| | |
|---|---|
| **Black-box instance: XXX (three copies of the same string)** | |

| **Observations:** | BBABBABABBABBABABBABBABA is generated by the black box |
|---|---|
| | BABBABBABBABBABBAB is generated by the black box |
| | ABBABBABB is generated by the black box |
| | ABAAAABABAAAABABAAAAB is generated by the black box |
| | AAA is generated by the black box |
| | AAAAAA is generated by the black box |
| | BBB is generated by the black box |
| | ABAABAABAABAABAABA is generated by the black box |
| | ABABAB is generated by the black box |
| | ABAABAABA is generated by the black box |
| | ... |
| | {More observations} |
| | ... |
| | BBBBAABABABBBBAABABABBBBAABABA is generated by the black box |
| | ABABAB is generated by the black box |
| | AAABAABAAABAABAAABAAB is generated by the black box |
| | ABABAB is generated by the black box |
| | BABBBBBBBABABBBBBBBBABABBBBBBBBA is generated by the black box |
| | ABBBABBBBABBB is generated by the black box |
| | BABABA is generated by the black box |
| | BBABAABBABBABABAABBABBABAABBA is generated by the black box |
| | AAAAAA is generated by the black box |

| **Model response: (Overlooking)** | The black box generates strings that are composed of repeated patterns or segments. Each string can be broken down into one or more segments, where each segment is a repetition of a smaller pattern. The patterns can be composed of the letters 'A' and 'B', and the repetition of these patterns can vary in length. Additionally, the black box can generate an empty string. |
|---|---|
| | The key characteristics of the generated strings are: |
| | 1. The strings can be empty. |
| | 2. The strings can be composed of repeated patterns of 'A' and 'B'. |
| | 3. The patterns themselves can vary in complexity and length. |
| | 4. The repetition of these patterns can be consistent or vary within the string. |
| | Examples of patterns include: |
| | - Single letters repeated (e.g., "AAA", "BBB"). |
| | - Simple alternating patterns (e.g., "ABABAB", "BABABA"). |
| | - More complex patterns with varying repetition (e.g., "ABBABBABB", "BABBABBAB-BABBABBAB"). |
| | **The rule can be estimated as: The black box generates strings by repeating patterns of 'A' and 'B', where the patterns can be simple or complex, and the repetition can be consistent or vary within the string.** |

Table 4: Overlooking example of Formal Language. LLM estimates a rule of the black-box instance without utilizing the observed data to come up with the hypothesis.

---

**Black-box instance: alpha=[0.1,0.1,0.4,0.4]; rho=0.4**

---

**Observations:**

```
Basket1:  [87.79948556831374, 39.184851241810016,
22.551465649464433, 42.84633678341351];
Basket2:  [66.20150946367784, 80.43281524767865,
68.02177295663886, 8.112415033884117];
Preference:  Basket1
Basket1:  [20.086807088533046, 79.79230548488334,
8.817190035426963, 90.22168650956948];
Basket2:  [90.40796323190197, 70.51080246059688,
69.96510352125082, 48.476064999077785];
Preference:  Basket2
Basket1:  [8.003392409139845, 35.23615132704936,
76.3120116750855, 59.18156984237496];
Basket2:  [31.278783818535036, 18.047644469275518,
21.85545800494433, 53.88293391636068];
Preference:  Basket1
```

...

{More observations}

...

```
Basket1:  [40.80654893485903, 14.795240036848078,
37.35074210377093, 79.44298024697972];
Basket2:  [51.30239830060435, 50.0604427932334,
21.59530896237239, 36.88204913376821];
Preference:  Basket1
Basket1:  [75.18968775093136, 42.30088453163497,
18.388385810882728, 95.92185012513752];
Basket2:  [93.37477559149177, 86.19812315165629,
49.438229718409715, 98.68538582332832];
Preference:  Basket2
Basket1:  [70.30309661179918, 94.48475506394722,
78.28267517818938, 70.90519145824096];
Basket2:  [63.648862012782175, 99.80230448540196,
61.33979160499423, 78.38413565179029];
Preference:  Basket1
```

---

**Model response:**
**(Overlooking)**

To reverse engineer the CES utility function, we need to determine the parameters $a_i$ and $\rho$ that best fit the observed preferences between baskets. The CES utility function is given by:

$$U(\mathbf{x}) = \left( \sum_{i=1}^{n} a_i \cdot x_i^\rho \right)^{1/\rho}$$

where $a_i$ are the weights for each good and $\rho$ is the elasticity parameter.
Steps to Determine Parameters:

- Analyze Preferences: For each pair of baskets, the basket with the higher utility is preferred.
- Estimate $\rho$: The parameter $\rho$ affects the degree of substitutability between goods. $\rho = 1$ is perfect substitutes, $\rho = 0$ is Cobb-Douglas, $\rho < 0$ is complementarity.
- Estimate $a_i$: The weights $a_i$ sum to 1 and reflect the importance of each good.
- Iterative Fitting: Use optimization to adjust $a_i$ and $\rho$ to minimize preference violations.

**Estimation: After analyzing the data and testing various combinations, the following provides a good fit for the observed preferences:**
Best Fit Utility Function:

$$U(\mathbf{x}) = \left( 0.3x_1^{0.5} + 0.2x_2^{0.5} + 0.3x_3^{0.5} + 0.2x_4^{0.5} \right)^2$$

This is derived by fitting the parameters to minimize preference violations. The weights $a_i$ reflect the relative importance of each good, and $\rho = 0.5$ is selected to match substitutability.

---

Table 5: Overlooking example of Math Equation. Instead of explicit plug-in all observed data into the utility function, LLM estimates $\alpha$ and $\rho$.

| Black-box: Formal Languages | |
| --- | --- |
| **Black-box instance:** | Strings over {A, B} where no 2 A's ever appear in a row |
| **Observations:** | BBBABBBAB is generated by the black box
" " is generated by the black box
BA is generated by the black box
ABABABABABB is generated by the black box
...
{more observations (included the active interaction)}
...
BBBBA is generated by the black box
ABABBA is generated by the black box
BBBBA is generated by the black box |
| **Model response:** | All **nonempty** strings over {A, B} that end with A and in which every run of consecutive B's has **length that is a multiple of 3** (i.e., B's occur only in blocks of length 3, 6, 9) |

Table 6: **GPT-5 Overcomplication Example.** GPT-5 overfits tiny patterns and invents very rigid structural rules. For example, "B blocks must be multiples of 3".

| Black-box: Formal Languages | |
| --- | --- |
| **Black-box in-stance:** | Strings over {A, B} where no 2 A's ever appear in a row |
| **Observations:** | BAB is generated by the black box
ABBA is generated by the black box
BABBABABBABBBABBBA is generated by the black box
...
{more observations}
...
ABBBABBABBBBBAB is generated by the black box
BBBBABA is generated by the black box
BAAB is not generated by the black box |
| **Model response:** | Strings over the alphabet {A, B} that contain at least one A and do not have three consecutive A's. In other words: The *empty string is not allowed. At least one A must appear.* The substring **AAA is forbidden**. All such and only such strings are generated. |

Table 7: **Claude 4 Overcomplication Example.** Claude-4 adds broad, grammar-like constraints with no evidence, for example, "no AAA" or "must contain an A").

| Black-box instance: alpha=[0.6, 0.4]; rho=1.0 | |
|---|---|
| **Observations:** | Basket1: [72.39463042444831, 98.20729968864211] |
| | Basket2: [84.11368818104853, 8.275148546306013] |
| | Preference: Basket1 |
| | |
| | Basket1: [62.30767597748462, 57.746856481974625] |
| | Basket2: [35.695267908460735, 98.66972480028333] |
| | Preference: Basket2 |
| | |
| | Basket1: [23.473671703790554, 25.832543390568617] |
| | Basket2: [94.42505284142774, 86.51364378020297] |
| | Preference: Basket2 |
| | ... |
| | {more observations} |
| | ... |
| | Basket1: [97.53278945423664, 56.50000818388516] |
| | Basket2: [21.198178991308247, 24.95045669114201] |
| | Preference: Basket1 |
| | |
| | Basket1: [25.125924557631045, 80.9410265539493] |
| | Basket2: [16.473104257940324, 57.09099238116093] |
| | Preference: Basket1 |
| | |
| | Basket1: [41.83258811801525, 78.58700060612355] |
| | Basket2: [51.231037581195686, 58.128202352337446] |
| | Preference: Basket1 |
| **Model response:** | We have 2 goods (since baskets are 2-vectors). The CES utility is $U(x) = (a_1 x_1^\rho + a_2 x_2^\rho)^{1/\rho}$, with $a_1, a_2 \geq 0$, $a_1 + a_2 = 1$ (to 1 decimal), and $\rho$ to 1 decimal. |
| | From observations: - When both goods are large in Basket 2 vs. modest in Basket 1, Basket 2 is preferred (third pair). - This implies $\rho > 0$ or at least not strongly negative. - Yet in pair 1, Basket1 beats Basket2 even though Basket2 has larger $x_1$ but very small $x_2$. - Overall pattern hints at moderate substitution. |
| | Func: $U(x_1, x_2) = (0.7 \times x_1^{-0.6} + 0.3 \times x_2^{-0.6})^{1/-0.6}$ |

Table 8: **GPT-5 Overlooking Example.** GPT-5 uses a part of observed data instead of the full set of observations to identify the parameters that the data clearly imply.

| **Black-box instance: alpha=[0.6, 0.4]; rho=1.0** | |
|---|---|
| **Claude 4** | |
| **Observations:** | Basket1: [58.01934185512253, 39.72973152295986]
Basket2: [37.64385409318356, 98.0711484140796]
Preference: Basket2

Basket1: [96.14209674372151, 19.312222198346394]
Basket2: [4.602811723537094, 3.142014475647384]
Preference: Basket1

Basket1: [80.27391244506391, 93.75994670581349]
Basket2:[13.26562127801874, 21.847074478777998]
Preference: Basket1

[more observations]

Basket1: [3.2554749911971794, 43.1362643647149]
Basket2: [11.048521718042805, 26.732069089490366]
Preference: Basket1

Basket1: [22.86209786031542, 65.04564626369694]
Basket2: [35.6791027722567, 18.851472251439098]
Preference: Basket1

Basket1: [50.86001401577884, 4.898492001384545]
Basket2: [10.991202877707694, 98.8352797235276]
Preference: Basket2 |
| **Model response:** | Based on the given observations and the structure of the CES utility function, I've analyzed the data to estimate the parameters. Here's my best estimate for the utility function:
$U(x) = (0.7 \times x_1^{0.9} + 0.3 \times x_2^{0.9})^{1/0.9}$
This function represents my best estimate for the CES utility function that fits the given observations.
The reasoning behind these estimates:
1. The weights (a_i) sum to 1 as required (0.7 + 0.3 = 1).
2. The preference for Basket1 in most cases where the first good has a higher quantity suggests a higher weight for the first good.
3. The rho value of 0.9 indicates a high degree of substitutability between the goods, but not perfect substitutes (which would be rho = 1).
4. This function satisfies most of the given observations. |

Table 9: **GPT-5 Overlooking Example.** Claude-4 jumps straight to a nice-looking observed data and assigns numbers that "feel plausible," without checking whether that function is consistent with the examples.

