# OpenReview forum: "Assessing the Reverse-Engineering Abilities of Large Language Models"
_ICLR.cc/2026/Conference — Submitted to ICLR 2026_

### Official Review · Reviewer_yYYx · 2025-10-29

**Soundness:** 2
**Presentation:** 2
**Contribution:** 3
**Rating:** 4
**Confidence:** 2

**Summary:**

This paper evaluates whether large language models (LLMs) can reverse-engineer unknown black-box problems, which are usually handled by Bayesian algorithms. The authors formalize this problem, curate the noise-free dataset from previous works and benchmark the selected LLMs (GPT-4o, in the main text) across three domains, including programs, formal languages, and mathematical equations and comparing passive observation versus active intervention paradigms. The authors identify two main failure modes, overcomplication and overlooking, appearing in the usual case without intervention and try to improve it with the proposed intervention. The evaluation is based on natural language reward/score (judged by another LLMs). Results show that LLMs can plateau under passive observation but show improvement with intervene across three experiments.

**Strengths:**

- Built a measurable reverse-engineering benchmark and evaluate the LLMs’ performance on it, which can be relevant to build autonomous research agents.
- Curated three setups from previous studies across formal language, math, and program for experiments and evaluations, which may be used as evaluation for future works

**Weaknesses:**

- Relies on LLMs (in specific GPT-4o in the experiments) as the nature language “judge” model, which may bias scoring and introduce circular evaluation effects compared to verifiable score or rewards.
- The obtained performance improvement under “intervention” might reflect prompt length or potential CoT-like reasoning help instead of hypothetically being contributed by actively collecting data. Necessary ablation studies on this is absent or not enough given Table 1.

**Questions:**

- The Bayesian inference baseline presumes full access/configuration to hypothesis space and need prior knowledge to define, does this set an unfair or unachievable upper bound for LLMs? If not, what is a more comparable Bayesian baseline to LLMs?
- Can you show with corresponding experimental evidence to justify the LLMs truly plan informative interventions and then make better “reverse-engineering” through intervention, instead of by (relatively trivial) increased reasoning length. In other words, what is the effect of reasoning verbosity (CoT length) versus actual hypothesis refinement?
- Can you also show the variance/std of the improvement margin in Fig 5?
- In Fig 8, it seems for quite a few (LLMs, Task) pair, the intervention does not help for specific data points used, as well as does not show consistent improvement along with increased data points. Could the authors explain around this phenomenon?

---

> ### Author Response · Authors · 2025-11-24
>
> We thank the reviewers for their detailed comments and suggestions.
>
> ## Weaknesses
> > Relies on LLMs (in specific GPT-4o in the experiments) as the nature language “judge” model, which may bias scoring and introduce circular evaluation effects compared to verifiable score or rewards.
>
> We want to clarify that GPT-4o does not introduce circular evaluation. In our study, the rubric is objective: a rule is either correct or not, and simplicity is only applied after correctness, which keeps the scoring stable. Also, LLM-as-judge is a common evaluation method, and prior work shows GPT-4–level models match or exceed human annotation quality [1, 2].
>
> As shown in Appendix G.1, we include judgments from GPT-o3, Claude 4, and human annotators. The agreement score suggests that all are within the "substantial agreement" range. Remaining variance mainly comes from the simplicity component, which does not change our conclusions.
>
> [1] G-Eval: NLG Evaluation using GPT-4 with Better Human Alignment, EMNLP 2023
>
> [2] LLMs-as-Judges: A Comprehensive Survey on LLM-based Evaluation Methods
>
> > The obtained performance improvement under “intervention” might reflect prompt length or potential CoT-like reasoning help instead of hypothetically being contributed by actively collecting data. Necessary ablation studies on this is absent or not enough given Table 1.
>
> > What is the effect of reasoning verbosity (CoT length) versus actual hypothesis refinement?
>
> As noted in Appendix E, the intervention prompt does not provide any structured reasoning template or step-by-step scaffold. The prompt only asks the model to think about what additional information would be useful for the next query. Thus, all reasoning is entirely free-form. Here is an example of how LM reason and design next query:
>
> *Black-box instance (Formal Language):* Strings over {A, B} where the number of A's is even
>
> *Observations:*
>
> ABABABBBABAA is generated by the black box
>
> ABAAABAAAAAAAABB is generated by the black box
>
> BABBBBBAAABBBBBABBBA is generated by the black box
>
> …
>
> AA is generated by the black box
>
> BBBBBBBBBBBBBBBBBBB is generated by the black box
>
> B is generated by the black box
>
> ABABAB is generated by the black box
>
> *Intervention with reasoning(n):*
>
> The strings vary in length, from as short as 1 character (B) to as long as 20 characters. The strings consist only of the characters 'A' and 'B'. There is no apparent fixed pattern of alternation between 'A' and 'B'. Some strings are composed entirely of 'A's or 'B's
> The black box seems to generate strings that consist of any combination of 'A's and 'B's. There doesn't appear to be a restriction on the order or the number of 'A's and 'B's, as long as the string is non-empty.
>
> To test this hypothesis, we should try a string that is a combination of 'A's and 'B's in a random order, which hasn't been explicitly tested yet. If the black box accepts this string, it would support the hypothesis that any combination of 'A's and 'B's is valid.
> Let's test the string "ABABA"
>
> *Black box’s response:*
>
> ABABA cannot be generated from black box.
>
> *Intervention with reasoning (n+1):*
>
> Based on the observations and the results of the tests, it seems that the black box does not generate all possible combinations of 'A's and 'B's.
>
> To refine our hypothesis, let's test a string that has a different pattern, such as even number of 'A', which has been accepted in previous tests. Let's try "ABABAB"
>
> **Explanation:**
>
> As you can see in this example, the reasoning is used to design the next query rather than interpreting the existing observations, and the improvement comes from acquiring new information. Also, we agree that thinking more can improve performance. This is expected: to choose a useful intervention, the model must first consider what it does not yet know and what question would reduce that uncertainty. This reasoning is part of the intervention process itself --- not the result of any hidden structure in the prompt and it does not appear in the passive learning.

---

> > ### Author Response · Authors · 2025-11-24
> >
> > ## Questions
> > > The Bayesian inference baseline presumes full access/configuration to hypothesis space and need prior knowledge to define, does this set an unfair or unachievable upper bound for LLMs? If not, what is a more comparable Bayesian baseline to LLMs?
> >
> > In our paper, we use the Bayesian baseline as a reference upper bound for the task. Given a fully specified hypothesis family and noise-free observations, it represents the best achievable performance under this setup. Also, this upper bound is not "unachievable" for LLMs in practice: LLMs also rely on their own prior knowledge. In our paper, we observe that in some cases they approach or even outperform this reference. For example, in Figure 3 (Formal Language), the intervention setting is already very close to the Bayesian baseline, and at $\(n_{\text{obs}}=10, n_{\text{intv}}=10\)$ even slightly better.
> >
> > > Can you also show the variance/std of the improvement margin in Fig 5?
> >
> > Yes, we updated the variance of the improvement margin in Fig 5 in the revised version.
> >
> >
> > > In Fig 8, it seems for quite a few (LLMs, Task) pair, the intervention does not help for specific data points used, as well as does not show consistent improvement along with increased data points. Could the authors explain around this phenomenon?
> >
> > We agree that intervention does not always improve performance. To clarify, we do not claim that intervention should always help. Our conclusion is that when observation no longer provides additional information, intervention can help. We observe that intervention is only effective when the model can ask informative questions. Some models fail to generate useful queries for certain tasks, so the extra interventions do not improve performance and can even destabilize it.

---

> > > ### Comment · Reviewer_yYYx · 2025-11-26
> > >
> > > Thanks the authors for their rebuttal response. I would like to maintain my score since it basically fits my assessment. I still have the following key concern remained: the use of GPT-4o as the assessment score may not provide a rigorous evaluation suite in my opinion. The LLM-based scoring can serve as surrogate but should be careful when giving quantitative evaluation especially the reverse engineering helper is also GPT-4o. Since the human annotation can be intractable, it would be more convincing to incorporate multiple LLMs as judge then simply showing the Cohen's kappa. That being said, I agree this work can be novel and relevant, so long as with better evaluation protocol.

---

> > > > ### Author Response · Authors · 2025-12-04
> > > >
> > > > Thank you for the response. We would like to clarify that our evaluation already includes agreement scores with Claude 4, o3, and human annotators, all showing substantial consistency with GPT-4o. These results are included in the submitted version (Appendix I) and were restated in our previous response. As shown in [1,2], such levels of agreement indicate that GPT-4o is sufficiently reliable as an evaluator.
> > > >
> > > > We agree that incorporating additional LLM judges could make the evaluation even more rigorous. However, this would not change our main conclusion: current LLMs are unreliable at recovering the underlying mechanism even in simple and controlled settings. In addition,  the goal of this paper is not to compare the judging abilities of different LLMs. Using multiple judges across all experiments would be prohibitively expensive, and since GPT-4o already shows strong and validated agreement with humans and other LLMs, our evaluation protocol is sufficiently rigorous to support our conclusions.
> > > >
> > > > [1] The Measurement of Observer Agreement for Categorical Data (1977)
> > > >
> > > > [2] Understanding Interobserver Agreement: The Kappa Statistic (2005)

---

### Official Review · Reviewer_xSfu · 2025-10-30

**Soundness:** 3
**Presentation:** 4
**Contribution:** 2
**Rating:** 4
**Confidence:** 4

**Summary:**

The paper evaluates the abilities of LLMs at forming hypotheses to explain the behavior of a black-box system given input/output observations (and the ability to query the system further). Through experiments on three tasks (python lambda expression, formal language rules, math equation inference), the authors find that (1) LLMs from frontier still lag behind Bayesian systems (oracle baseline), (2) through an expert human evaluation, the failures can be commonly attributed to either overly complicated rules, or ignoring/overlooking some information in the examples, (3) when allowing the LLM to interact with the system, performance improves most commonly when the the intervention is unconstrained and the LLM can reason from past examples (as opposed to verbalizing the black box as python code).

**Strengths:**

1) The task is simple and well motivated, the experiments are well designed to evaluate model behavior and then analyze failure cases and the writing is clear and easy to follow.

2) The extension from pure observation-based to interaction-based reverse engineering is intuitive, and the experiments showing that the active experience does not transfer to other models (Sec 5.3) is compelling as it implies that the reverse engineering process is unique to the particular model.

3) The manual analysis is informative in categorizing model behavior, and allows us to ask more qualitative questions to go beyond individual benchmark numbers (see Q2 and 7 below).

**Weaknesses:**

One result that is important for verifying the claim is the variance in performance across different sets of observations (of the same size). What do you mean by variance by seeds in L.234? Is it the generation hyperparameter, or also the sampling algorithm for the particular observations used for reverse engineering?

This could potentially be in Questions for the authors, but I think it's worth bearing mention that the work could do a better job engaging other work, particularly published literature that tries to verbalize the decision functions for labeling in ICL examples [1], describe datasets and distributions in text [3] and elicit preferences of users in natural language [2, 4]. The tasks are different, but the principles of the experiment seem similar. This would be fine, but I bring it up here as the paper lacks a strong, surprising finding beyond our priors of LLMs from the literature. To mitigate this, I think moving from a single amorphous 'LLM' category to analysis between models could help. A potential finding that is already in the results is the relative performance of different models, which got pushed to Appendix F. Do effects such as scale, or training recipe, or model family affect performance? Are models well calibrated in their rule judgments?

[1] Si, Chenglei, et al. "Measuring Inductive Biases of In-Context Learning with Underspecified Demonstrations." The 61st Annual Meeting Of The Association For Computational Linguistics. 2023.

[2] Li, Belinda Z., et al. "Eliciting Human Preferences with Language Models." The Thirteenth International Conference on Learning Representations.

[3] Zhong, Ruiqi, et al. "Goal-driven discovery of distributional differences via language descriptions." Advances in Neural Information Processing Systems 36 (2023): 40204-40237.

[4] Handa, Kunal, et al. "Bayesian preference elicitation with language models." arXiv preprint arXiv:2403.05534 (2024).

**Questions:**

1) One concern is that the experiments report performance over multiple seeds. Do you also vary the particular examples given to the model in your experiments? Is N=10 a proper subset of the examples in N=50? What is the variation as a

2) For figure 8, is the judge always GPT-4o? Do the different models have similar or very unique interaction patterns, in terms of the examples they seek clarification about? Do they tend have have different profiles among constructing new queries, response pairs, and concluding the analysis?

3) L.236-237- When you say that the "LLM performs M = {5, 10, 20, 50} rounds of interventions conditioned on the initial set of 10 observations", I took the points on the graph to mean 10 + $m \in M$ turns. Is this correct? Does the interaction baseline performance change if you provide more initial observations?

4) Could you comment on issues of potential data leakage of the rules (to be inferred) into the model pre-training? From [1], LLMs are able to decipher certain algorithms more than others due to prevalence in pre-training.

5) Given that the experiments with other models are in the appendix, why present GPT-4o in the main paper when it is the same as the judge model?

6) I think the absence of a non-oracle baseline is making it hard to interpret some of the results (since the Bayesian methods have a set of rules to search within, L.882). This could be where something like comparing the performance of different models on one graph (either of model families or scales or training steps) could help.

7) Do different models fail systematically at certain examples (i.e. longer or more complex ones)? Do different models overcomplicate the same examples?

[1] Prabhakar, Akshara, Thomas L. Griffiths, and R. McCoy. "Deciphering the Factors Influencing the Efficacy of Chain-of-Thought: Probability, Memorization, and Noisy Reasoning." Findings of the Association for Computational Linguistics: EMNLP 2024. 2024.

---

> ### Author Response · Authors · 2025-11-24
>
> We thank the reviewer for acknowledging our experimental design and in-depth analyses, and for providing valuable feedback. We address each point in detail below.
>
> > “What do you mean by variance by seeds in L.234? Is it the generation hyperparameter, or also the sampling algorithm for the particular observations used for reverse engineering?”
>
> > One concern is that the experiments report performance over multiple seeds. Do you also vary the particular examples given to the model in your experiments? Is N=10 a proper subset of the examples in N=50?
> In our experiments, a seed corresponds to one black-box instance (Program, Language, or Math Equation). All observations for that instance are generated deterministically, so the variance across seeds reflects differences between black-box instances, not sampling randomness or decoding randomness.
>
> For each instance, the observation sets are nested: N=10 is the first 10 observations of the N=50 set. We do not re-sample observations for different N. This ensures that changes across N come from having more evidence about the same underlying instance rather than sampling noise.
>
>
> > the work could do a better job engaging other work [...] the paper lacks a strong, surprising finding beyond our priors of LLMs from the literature.
>
> Thank you for providing these references. First, to clarify, our paper’s main point is to examine whether LLMs can reliably recover an explicit, fully checkable mechanistic rule of a black-box system from observations and interventions.
>
> To organize these previous results that you cited and explain the differences, we created a table below of the previous work. We have added them to the related work in the revised paper.
>
>
> | Paper | Inductive reasoning (passive learning) | Active intervention (active learning) |
> |-------|----------------------------------------|----------------------------------------|
> | [1] | yes | no |
> | [2] | yes | no |
> | [3] | yes | no |
> | [4] | yes | no |
>
> As shown in the Table, none of them evaluates whether LLM can recover the exact underlying mechanism of a black box system through active intervention. Our novel contributions and findings are:
>
> - We provide a fully controllable evaluation protocol to test whether LMs can reverse-engineer black-box systems from both passive observations and active interventions.
>
> - Our key insights are that 1) LMs still struggle to reliably reverse-engineer a black-box system from only observed data, and 2) effective intervention can help LMs mitigate the failure modes of overcomplication and overlooking.
>
> [1] Si, Chenglei, et al. "Measuring Inductive Biases of In-Context Learning with Underspecified Demonstrations." The 61st Annual Meeting Of The Association For Computational Linguistics. 2023.
>
> [2] Li, Belinda Z., et al. "Eliciting Human Preferences with Language Models." The Thirteenth International Conference on Learning Representations.
>
> [3] Zhong, Ruiqi, et al. "Goal-driven discovery of distributional differences via language descriptions." Advances in Neural Information Processing Systems 36 (2023): 40204-40237.
>
> [4] Handa, Kunal, et al. "Bayesian preference elicitation with language models." arXiv preprint arXiv:2403.05534 (2024).
>
> > moving from a single amorphous 'LLM' category to analysis between models could help.
>
> As noted in Figure 8, we compare different LMs. The results suggest a general trend that when additional observations no longer improve a model's understanding of the black box, active intervention can still help. We make this point clearer in the revised version.
>
> > A potential finding that is already in the results is the relative performance of different models, which got pushed to Appendix F.
>
> Thank you for the suggestion. In the revised version, we move the comparison of different models from Appendix F into the main text.
>
>
> > “Do effects such as scale, or training recipe, or model family affect performance? Are models well calibrated in their rule judgments?”
>
> We agree that analyzing model differences is useful. In the revised version, we add additional failure examples for different models in the Appendix. We observe that GPT-5 often adds very narrow structural conditions that are not supported by the data, while Claude-4 tends to add broad constraints that also do not appear in the observations. These suggest that LLMs are not a single homogeneous category and that different models exhibit distinct error patterns.

---

> > ### Author Response · Authors · 2025-11-24
> >
> > ## Questions
> > >  For Figure 8, is the judge always GPT-4o?
> >
> > Yes, the judge is always GPT-4o. We have clarified this in the revised paper.
> >
> > > Do the different models have similar or very unique interaction patterns, in terms of the examples they seek clarification about? Do they tend have have different profiles among constructing new queries, response pairs, and concluding the analysis?
> >
> > In our paper, we do not observe a clear systematic difference across models. From the qualitative analysis, we observe that some models generate more exploratory queries while others repeat similar queries or conclude earlier, but these patterns are not consistent enough to support strong conclusions. We note this as a future direction in the revised version.
> >
> > > L.236-237- When you say that the "LLM performs M = {5, 10, 20, 50} rounds of interventions conditioned on the initial set of 10 observations", I took the points on the graph to mean 10 +  turns. Is this correct?
> >
> > Yes, that is correct.
> >
> > > Does the interaction baseline performance change if you provide more initial observations?
> >
> > In our experiments, the number of initial observations is a fixed design choice made early on. In the early stage of this paper, we find that performance in the observation-only setting typically plateaus after around 10 observations, suggesting that additional observations do not provide much new information. Setting at 10 allows us to attribute later improvements to the interventions rather than to the initial data size.
> > To provide more details, we ran an additional test on the Language task with 20 initial observations and 10 interventions, which achieved a score of 31.2. For comparison, using 10 initial observations and 20 interventions achieves 34.3.
> > To clarify, we do not claim that 10 is optimal, it is a tunable parameter. We use 10 based on the observation-only plateau and keep it fixed for consistency in our experimental design.
> >
> >
> > > Could the black-box rules or mechanisms appear in pretraining data, influencing results?
> >
> > Thanks for raising this point. We believe pretraining leakage has minimal impact. All rules in our tasks are synthetically generated, and the exact rules do not appear in pretraining corpora. While LMs may have been exposed to conceptually similar ideas, this does not inflate performance: the models still show clear overcomplication and overlooking failures, indicating that their behavior is not driven by retrieving memorized rules.
> >
> > > Given that the experiments with other models are in the appendix, why present GPT-4o in the main paper when it is the same as the judge model?
> >
> > We present GPT-4o in the main paper because we run the experiments with three seeds and compare directly to the Bayesian baseline. Due to cost limits, we cannot run all models at this scale, so the remaining results are reported in the appendix. Using GPT-4o as a judge is simply an experimental design choice.
> >
> >
> > > I think the absence of a non-oracle baseline is making it hard to interpret some of the results (since the Bayesian methods have a set of rules to search within, L.882).
> >
> > We clarify that the Bayesian system is not used as an oracle baseline. It is a reference method with a fixed hypothesis space, included to indicate the difficulty of each black-box instance rather than to represent an upper bound of model performance.
> >
> > > This could be where something like comparing the performance of different models on one graph (either of model families or scales or training steps) could help.
> >
> > In our paper, all the models are closed-source, except for Llama-3-70B, and we do not have access to their training stages or scale information, so comparisons of "scale" or "training step" are impossible at this time. In addition, our figures are designed to highlight the gap between observation-only and observation-intervention conditions. Plotting all models together would dilute this contrast and obscure the effect of active interventions, which is the core focus of the paper.
> >
> > > Do different models fail systematically at certain examples (i.e. longer or more complex ones)? Do different models overcomplicate the same examples?
> >
> > Yes, different models do fail on the same examples, but their failure patterns differ. We have added this analysis in the Appendix G.2.

---

> > > ### Comment · Reviewer_xSfu · 2025-11-26
> > > **Rebuttal acknowledgment**
> > >
> > > Thank you for engaging with the review feedback! I'm open to updating my score, but still remain unconvinced/unclear about specific points which, taken together, are why I so far retain my existing score:
> > > - On the reliability of GPT-4o (Q5): In Appendix I, the evaluation of GPT-4o is done with 30 examples (10 with each type) and the correlation with human and reasoning models is shown to be high. My concerns with this are (a) Sample size - 10 per task seems very low, how do you confirm that this is representative?, (b) How many human annotations were done per example? And how do you ensure that these human annotations were reliable? These details are missing in the draft. (c) The examples seem to be reference generations. The concern that I had in Q5 was mainly that GPT-4o might be reliable at evaluating the generations of other models, while also being unreliable for GPT-4o generations (as informed by prior work). That's why I question highlighting the GPT-4o generations.
> > > - On a non-oracle baseline (Q6): Yes, perhaps 'oracle' was the wrong choice of wording. But given that there is a fixed reference space that definitely contains the right answer, isn't it still a much easier task for this method? It feels like a slightly different problem to compare a generation method to a selection as the main baseline.
> > > - On comparison to literature: From the summary, the contribution feels commensurate with my score. If the interactivity/ability to query for examples is the key contribution, that seems more a feature of LLMs themselves than the evaluation protocol, particularly when the unconstrained exploration is what does well (no prompt scaffolding etc.). So, while the analysis is clear and interesting, it feels like a 'natural thing to try' based on what we know already. Perhaps the right way to frame my question is to ask what you would present as the most surprising finding to the research community?

---

> > > > ### Author Response · Authors · 2025-12-04
> > > >
> > > > We thank reviewer's response, we provide more detailed information about the reliability of using GPT-4o as a judge and our main contributions and insights below.
> > > >
> > > > ##  Reliability of GPT-4o as a Judge
> > > >
> > > > Thank you for raising the question about the validity of using an LLM judge (GPT-4o) and for further clarifying Q5. Below we provide a brief recap of our LLM-judge setup and evidence supporting its reliability.
> > > >
> > > > Our LLM judge receives (i) a rubric consisting of criteria that sum to 10 points (rescaled to 100 in the paper) and (ii) the ground-truth description of the black-box instance. In our paper, similar to [1], the rubric is fully objective: each rule is either correctly recovered or not. The simplicity criterion is applied only after correctness is assessed, ensuring that the scoring remains stable and consistent. Since the evaluation is rule-based rather than preference-based, the scoring is not influenced by model bias.
> > > >
> > > > > a) Sample size - 10 per task seems very low, how do you confirm that this is representative?
> > > >
> > > > We randomly sampled 30 instances from the full set of 196 (50 math equations + 46 languages + 100 programs). Earlier studies on kappa agreement show that small sample sizes of this scale are considered adequate and representative for reliability checks [2, 3]. Therefore, using 30 instances overall is sufficient for validating the reliability of using GPT-4o as the judge. As the results, we obtain Weighted Cohen's Kappa scores of 0.773 (Human), 0.752 (Claude 4), and 0.734 (o3). All of these fall within the "substantial agreement" range [4, 5].
> > > >
> > > > > b) How many human annotations were done per example? And how do you ensure that these human annotations were reliable? These details are missing in the draft.
> > > >
> > > > We follow the standard two-annotator setup used in [3]: we have two human annotators independently judge all sampled examples using the same rubric as GPT-4o, and we compute the agreement score by aggregating their two human scores. Thank you for pointing this out, we have added this clarification to Appendix I in the revised paper.
> > > >
> > > >
> > > > > c) The examples seem to be reference generations.
> > > >
> > > > We don't necessarily agree with this point since our evaluation is objective instead of subjective. Also, as shown in Appendix I, GPT-4o's scores have substantial agreement with human annotators, Claude 4, and o3 when evaluating GPT-4o's generations, suggesting that there is no systematic self-evaluation bias.
> > > >
> > > > Overall, we think using GPT-4o as a judge is reliable and we provide solid evidence to support that in our paper. Also, the use of LLM-as-Judge has been a common practice to evaluate model generation and GPT-4 level models have been shown to match or exceed human annotation in quality [6, 7] for *evaluating* generated text.
> > > >
> > > > [1] Prometheus: Inducing Fine-grained Evaluation Capability in Language Models
> > > > [2] Guidelines of the minimum sample size requirements for Cohen’s Kappa (2017)
> > > > [3] Interrater reliability: the kappa statistic (2012)
> > > > [4] The Measurement of Observer Agreement for Categorical Data (1977)
> > > > [5] Understanding Interobserver Agreement: The Kappa Statistic (2005)
> > > > [6] G-Eval: NLG Evaluation using GPT-4 with Better Human Alignment (2023)
> > > > [7] LLMs-as-Judges: A Comprehensive Survey on LLM-based Evaluation Methods (2024)
> > > >
> > > >
> > > > ## Using Bayesian Model as the Reference
> > > > > But given that there is a fixed reference space that definitely contains the right answer, isn't it still a much easier task for this method? It feels like a slightly different problem to compare a generation method to a selection as the main baseline.
> > > >
> > > > In our paper, the Bayesian baseline is not intended to compete with LLMs or act as an oracle. It provides a clear reference for how informative the observations are in a fully specified hypothesis space. We agree the tasks are slightly different, but this does not create an unfair comparison in our paper, because we use it only as an upper bound on how fully the observed data can be utilized, rather than as a direct competitor in performance.
> > > >
> > > >
> > > > ## Highlighting Our Main Contributions and Insights
> > > > > what you would present as the most surprising finding to the research community?
> > > >
> > > > To clarify, our focus has been on evaluating whether LLMs can reliably perform core steps in scientific processes, such as generating hypotheses from observed data and designing experiments to test and refine them. We use black-box reverse-engineering tasks as a representative testbed for this evaluation. Failure even in these simple settings raises concerns about their reliability in basic scientific reasoning. The key insight for the research community is that, despite recent advances in applying LLMs to science, we still cannot rely on them to consistently uncover the underlying mechanisms of a simple black-box system in a controlled setting.

---

### Official Review · Reviewer_sTYg · 2025-10-31

**Soundness:** 3
**Presentation:** 3
**Contribution:** 3
**Rating:** 6
**Confidence:** 3

**Summary:**

The paper tries to find out how well LLMs can reverse-engineer 3 black box systems: program, math and formal languages. They run three kinds of experiments: giving the LLMs only a randomly sampled set of observations, giving them the power to intervene and interact with the black box systems and allowing an LLM to see the observations obtained from intervention of another LLM. These results are compared to Bayesian systems — and show that LLMs lag behind them, especially when not allowed to intervene. The paper also identifies two failure models: over complication and over simplification which leads to errors in hypothesis generation.

**Strengths:**

- The paper is well written and easy to follow with helpful illustrations.

- The experiments are rigorous. The authors use 3 distinct tasks (program, formal languages and mathematical equations) along with a graded difficulty level to provide a fine-grained evaluation of model's capabilities. The evaluation spans six models with proper statistical analysis of the results.

- The paper focuses on the timely topic of assessing AI models as autonomous researchers for scientific discovery. Given these models are increasingly being used for research workflows, this work provides an important step forward.

**Weaknesses:**

- The LLM as a Judge paradigm here uses only 1 model with a 0-10 scale and although there’s agreement with humans, there isn’t 100% agreement. The prompt for judge is also slightly subjective which can lead to noisy results. Along with that, using a judge model from the same family as the one being analysis has been proven to be noisy and unreliable. How do you think you can address this concern in the paper?

- The two types of failure cases mentioned are very intuitive — irrespective of a model or human, one can only overcomplicate or oversimplify a given problem. It will be great if authors can add more concrete examples and analysis for the same, this would improve the readability of the paper.

- It will be really nice if authors can think of qualitative analysis on understanding the failure modes more closely and write about it in the paper. This will help.

- The experiments regarding transfer are not very rigorous and require more information to ensure we can count it as a significant contribution.

- More analysis of the limitations of the models would be helpful. See Questions section for more details.

- It is expected that the model will perform well when given more information, the result that it performs better in the observation-intervention setting compared to observation-only setting seems may be considered trivial. An analysis of how interventions differ from additional passive data would strengthen the paper.

**Questions:**

- Why does the performance usually plateau after 10 observations?

- In Line 412, it is said that in the experiments the LLMs are allowed to reason once every five queries. This needs to be explained better because some of the strategies inherently require reasoning in each step or before generating each query. Can authors add more details here.

- Can you share any examples where the models dynamically refine their hypothesis in response to their own interventions. It would help better understand why repurposing LLM's intervention data for another model does not work well.

- What intervention method was used for reporting results in Figure 3 and 4?

---

> ### Author Response · Authors · 2025-11-24
>
> We thank the reviewer for recognizing the value of our evaluation on AI models as autonomous researchers for scientific discovery and believe this is a timely and important research direction.
>
> ## Reliability and Accuracy of Using GPT-4o as a Judge
> To clarify, our evaluation is objective: a hypothesis is either correct or incorrect under the rubric. The rubric assesses (1) correctness and (2) simplicity, where simplicity is applied only after correctness is satisfied. This ensures stable scoring. The use of LLM-as-Judge is also standard practice for evaluating generated text, and GPT-4–level models have been shown to match or exceed human annotation quality [1, 2].
>
> As shown in the paper, we include evaluation results from GPT-o3, Claude 4, and human annotators. We sample 30 cases (10 from each black-box type) and compute Weighted Cohen’s Kappa for ordinal ratings. The agreement scores are: Human = 0.773, Claude 4 = 0.752, and o3 = 0.734. Although there is not 100% agreement, these values indicate substantial agreement (Landis & Koch, 1977). Most of the remaining variability comes from the simplicity component, which does not affect our conclusions.
>
> These already suggest the stability and reliability of using GPT-4o as judged in our experiments. We agree that using additional strong LMs as judges could further stabilize the numerical results. However, this is difficult to scale under cost constraints.
>
>
> [1] G-Eval: NLG Evaluation using GPT-4 with Better Human Alignment, EMNLP 2023
>
> [2] LLMs-as-Judges: A Comprehensive Survey on LLM-based Evaluation Methods
>
>
> ##  Qualitative analysis for Failure Modes
> > It will be really nice if authors can think of qualitative analysis on understanding the failure modes more closely and write about it in the paper.
>
> We agree that these two failure types are intuitive. As noted in Appendix G.1 in the submitted version, the failure modes come directly from human annotation of low-scoring cases.
>
> We provide quantitative analysis for each black-box type in Appendix G.2 in the submitted version. As you suggested, we add analysis in our revised version in Appendix F.2. In Program, the model often introduces an extra operation that never appears in the observations, producing a rule that is more complex than the true black-box behavior. In Formal Language and Math Equation, the model frequently ignores key evidence in the observations and outputs a hypothesis that does not use the available information.
>
> We also provide more examples for overcomplication and overlooking from different models (GPT-5 and Claude 4) below and in the Appendix of the revised paper (see Appendix G2):
>
> **overcomplication**
>
> Black-box: Formal Languages
>
> Black-box instance: Strings over {A, B} where no 2 A's ever appear in a row
>
> **GPT-5**
>
> *Observations:*
>
> BBBABBBAB is generated by the black box
>
> “” is generated by the black box
>
> BA is generated by the black box
>
> ABABABABABB is generated by the black box
>
> [more observations (included the active interaction)]
>
> BBBBA is generated by the black box
>
> ABABBA is generated by the black box
>
> BBBBA is generated by the black box
>
> *Model response:*
>
> All **nonempty** strings over {A, B} that end with A and in which every run of consecutive B’s has **length that is a multiple of 3** (i.e., B’s occur only in blocks of length 3, 6, 9)
>
> **Claude 4**
>
> *Observations:*
>
> BAB is generated by the black box
>
> ABBA is generated by the black box
>
> BABBABABBABBBABBBA is generated by the black box
>
> [more observations]
>
> ABBBABBABBBBAB is generated by the black box
>
> BBBBABA is generated by the black box
>
> BAAB is not generated by the black box
>
> *Model response:*
>
> Strings over the alphabet {A, B} that contain at least one A and do not have three consecutive A's. In other words: The *empty string is not allowed*. *At least one A must appear.* The substring **AAA is forbidden**. All such and only such strings are generated.
>
> **Explanation:**
> Both GPT-5 and Claude-4 overcomplicate; however, they fail in different "styles": GPT-5 makes the rule too narrow and technical, Claude-4 makes it too broad and semantic.
> More specifically, GPT-5 overfits tiny patterns and invents very rigid structural rules (like "B blocks must be multiples of 3") and Claude-4 instead adds broad, grammar-like constraints with no evidence (like “no AAA” or “must contain an A”).
>
> (*Continued on next comment*)

---

> > ### Author Response · Authors · 2025-11-24
> >
> > **overlooking**
> >
> > Black-box instance: alpha=[0.6, 0.4]; rho=1.0
> >
> > **GPT-5**
> >
> > *Observations:*
> >
> > Basket1: [72.39463042444831, 98.20729968864211]
> >
> > Basket2: [84.11368818104853, 8.275148546306013]
> >
> > Preference: Basket1
> >
> > Basket1: [62.30767597748462, 57.746856481974625]
> >
> > Basket2: [35.695267908460735, 98.66972480028333]
> >
> > Preference: Basket2
> >
> > Basket1: [23.473671703790554, 25.832543390568617]
> >
> > Basket2: [94.42505284142774, 86.51364378020297]
> >
> > Preference: Basket2
> >
> > [more observations]
> >
> > Basket1: [97.53278945423664, 56.50000818388516]
> >
> > Basket2: [21.198178991308247, 24.95045669114201]
> >
> > Preference: Basket1
> >
> > Basket1: [25.125924557631045, 80.9410265539493]
> >
> > Basket2: [16.473104257940324, 57.09099238116093]
> >
> > Preference: Basket1
> >
> > Basket1: [41.83258811801525, 78.58700060612355]
> >
> > Basket2: [51.231037581195686, 58.128202352337446]
> >
> > Preference: Basket1
> >
> > *Model response:*
> >
> > We have 2 goods (since baskets are 2-vectors).
> > The CES utility is
> >
> > $$
> > U(x) = (a_1 x_1^{\rho} + a_2 x_2^{\rho})^{1/\rho},
> > $$
> >
> > with $\(a_1, a_2 \ge 0\), \(a_1 + a_2 = 1\)$ (to one decimal), and $\(\rho\)$ to one decimal.
> >
> > From the observations:
> >
> > - When both goods are large in Basket 2 versus modest in Basket 1 (third pair), Basket 2 is preferred.
> >   This implies $\(\rho > 0\)$ (greater substitutability) or at least not strongly negative (which would make the utility closer to Leontief and emphasize balanced minima).
> >
> > - In pair 1, Basket 1 is preferred even though Basket 2 has a larger $\(x_1\) (84 > 72)$ but a very small $\(x_2\) (8 vs. 98)$.
> >   This suggests some complementarity $(\(\rho < 1\))$ because a shortage in one good lowers utility more.
> >
> > - Overall, the pattern points to moderate substitution: not Leontief-like, not perfect substitutes.
> >
> >
> > ```Func
> >
> > U(x1, x2) = (0.7 * x1^(-0.6) + 0.3 * x2^(-0.6))^(1 / -0.6)
> >
> > ```
> >
> >
> >
> > ** Claude 4 **
> >
> > *Observations:*
> >
> > Basket1: [58.01934185512253, 39.72973152295986]
> >
> > Basket2: [37.64385409318356, 98.0711484140796]
> >
> > Preference: Basket2
> >
> > Basket1: [96.14209674372151, 19.312222198346394]
> >
> > Basket2: [4.602811723537094, 3.142014475647384]
> >
> > Preference: Basket1
> >
> > Basket1:  [80.27391244506391, 93.75994670581349]
> >
> > Basket2:[13.26562127801874, 21.847074478777998]
> >
> > Preference: Basket1
> >
> > [more observations]
> >
> > Basket1: [3.2554749911971794, 43.1362643647149]
> >
> > Basket2: [11.048521718042805, 26.732069089490366]
> >
> > Preference: Basket1
> >
> > Basket1: [22.86209786031542, 65.04564626369694]
> >
> > Basket2: [35.6791027722567, 18.851472251439098]
> >
> > Preference: Basket1
> >
> > Basket1: [50.86001401577884, 4.898492001384545]
> >
> > Basket2: [10.991202877707694, 98.8352797235276]
> >
> > Preference: Basket2
> >
> > *Model response:*
> >
> > Based on the given observations and the structure of the CES utility function, I've analyzed the data to estimate the parameters. Here's my best estimate for the utility function:
> >
> > ```Func
> >
> > U(x) = (0.7 * x_1^0.9 + 0.3 * x_2^0.9)^(1/0.9)
> >
> > ```
> >
> > This function represents my best estimate for the CES utility function that fits the given observations.
> > The reasoning behind these estimates:
> >
> > 1. The weights $(a_i)$ sum to 1 as required $(0.7 + 0.3 = 1)$.
> >
> > 2. The preference for Basket1 in most cases where the first good has a higher quantity suggests a higher weight for the first good.
> >
> > 3. The rho value of 0.9 indicates a high degree of substitutability between the goods, but not perfect substitutes (which would be $rho = 1$).
> >
> > 4. This function satisfies most of the given observations.
> >
> > **Explanation**
> >
> > As you can see, GPT-5's problem is that it uses a part of observed data instead of the full set of observations to identify the parameters that the data clearly imply. Claude-4, in contrast, jumps straight to a nice-looking CES formula and assigns numbers that "feel plausible," without checking whether that function is consistent with the examples. Overall, GPT-5 overlooks by not doing enough inference, while Claude-4 overlooks by making up an answer without validating it.
> >
> >
> >
> > ## The Goal of Transfer Experiments
> > > The experiments regarding transfer are not very rigorous and require more information to ensure we can count it as a significant contribution
> >
> > The main goal of this analysis is to study whether models that cannot generate useful queries on their own can still benefit when given GPT-4o's intervention data. Therefore, this transfer experiment is not part of the main results, which is why we placed it in the analysis section rather than the core evaluation. We are happy to explore this direction further in future work, and we have added this point to the future directions section in the revised version in Lines 425-427.

---

> > > ### Author Response · Authors · 2025-11-24
> > >
> > > ## Questions
> > > > Why does the performance usually plateau after 10 observations?
> > >
> > > We observe that the performance plateaus after about 10 observations because the models do not reliably extract or integrate additional structure from more data beyond that point, even though more information is available. For example, in the Formal Language task with $L = {AB}^n$, an observation-only setting with $\(n_{\text{obs}} =  20$ produces 20 random strings such as AB or ABABAB, but the model does not consistently use the extra variation. In contrast, intervention allows the model to directly test targeted queries (e.g., asking whether ABABAB is valid), which provides more informative evidence than additional passive samples.
> > >
> > > >  In Line 412, it is said that in the experiments the LLMs are allowed to reason once every five queries. This needs to be explained better because some of the strategies inherently require reasoning in each step or before generating each query. Can authors add more details here.
> > >
> > > We use reasoning every five queries because our early trials showed that reasoning after every query caused the model to over focus on the most recent query–response pair. Since these parameters need to be fixed at the start of the project, we set it to 5.
> > >
> > > To provide more detail, we ran an additional comparison on the Language black box using GPT-5 with $\(n_{\text{obs}} = 10\)$ and $\(n_{\text{intv}} = 10\)$. Reasoning every five queries gives a score of 46.7, while reasoning after every query gives 40.9.
> > > However, this setting is not intended to be optimal. It is a tunable parameter, and we choose 5 based on a balance of performance, cost, and runtime. We clarify this in the revised version on page 8 (see footnote).
> > >
> > > > Can you share any examples where the models dynamically refine their hypothesis in response to their own interventions. It would help better understand why repurposing LLM's intervention data for another model does not work well.
> > >
> > > We provide an example of GPT-4o's intervention process in the Language task (shown below, and added to the Appendix in the revised paper). In this case, the model forms a hypothesis about the pattern and then queries the black box with an unseen string of a different length to test its own uncertainty.
> > >
> > > Language: $A^(2^n)$
> > >
> > > *Interventions:*
> > >
> > > Based on the observations provided, we can see that the black box generates strings consisting solely of the character 'A'. The strings that have been generated by the black box vary in length, but they all share the common characteristic of being composed entirely of the letter 'A'.
> > >
> > > ...
> > >
> > > From these observations, it seems that the black box generates strings of 'A's of any length.
> > > To further test this hypothesis, I will query the black box with a string of 'A's of a length that has not been observed yet: AAA
> > >
> > > *Black box’s response:* AAA cannot be generated by black box.
> > >
> > >
> > > > What intervention method was used for reporting results in Figure 3 and 4?
> > >
> > > Thank you for pointing out this. We use the Analyze-then-Query intervention in Figure 3 and 4, and we have clarified this in the revised version.

---

### Official Review · Reviewer_mbzm · 2025-11-01

**Soundness:** 2
**Presentation:** 3
**Contribution:** 2
**Rating:** 4
**Confidence:** 2

**Summary:**

This paper explores the abilities of LLMs to reverse engineer black-box systems. Three different types of black-box systems are explored: list-mapping programs, formal languages, and inferring the parameters of an economics math equation. Models are evaluated under three different conditions: a passive-only condition where they only receive observations from the program; an intervention condition where they're allowed to query the program; and an intervention-yoked condition, which is similar to the passive condition except that some of the observations are queries that were produced by another LLM. The failure modes of models are broken down into two different types: namely, overcomplication, which is inferring additional behavior that isn't supported by the data, and overlooking, where the LLM does not make full use of the available data.

LLMs behave below an ideal Bayesian observer in the passive-only condition, but the ability to actively query and intervene on the system recovers some, but not all, of that performance gap. Moreover, this seems to be tied to the process of active exploration itself, and not simply the informativeness of the examples, as the intervention-yoked condition does not produce the same performance.

**Strengths:**

- Originality
    - Weakness: I'm not terribly familiar with the active inference literature, but I am familiar with tasks of the type, LLM evaluations, that require LLMs to reverse engineer black-box systems. And it doesn't seem to me that either the tasks or the methodology used in this paper represent meaningfully novel contributions.
    - Strength: At the same time, I could simply be misinformed about the state of the literature.  It's possible that everyone is trying to jump the shark with end-to-end scientist AI and not doing careful analysis of each of the cognitive components, in which case this paper would represent a good grounding in the fundamentals.
- Quality
    - Weakness: It seems like only three attempts per task were performed. At least that's how I'm interpreting the phrase “report performance over three seeds.” This seems like a very low number of attempts. Granted, there are multiple “variants” of each task, corresponding to different numbers of data points, but only three runs on each task variant seems very low.
    - Strength: I was surprised by the use of LLMs as a judge, given that these seem like tasks with closed-form solutions that can be algorithmically verified, but the appendix showing its agreement with human annotators was reassuring.
        - Weakness: However, I don't know how to interpret the descriptiveness score. It's not clear to me how moving from a descriptiveness score of, for example, 30 to 50 would affect the model's ability to make use of it’s understanding of the system to do useful work.
    - Weakness: The LLM transfer experiment feels distracting. I would not expect to see improved performance when transferring results from one model to another if we don't see improved performance within a model. It feels like the paper would have been tighter and more cohesive by omitting that experiment.
- Clarity
    - Strength: The graphs were clearly presented, clearly labeled, and very readable.
    - Strength: Inclusion of example programs and agent responses was helpful.
    - Weakness: The diagram at the start of the paper was not great; it was a bit busy and not terribly informative. It did not do much to improve my understanding of overcomplication and overlooking, for example.
    - Weakness: There are four different intervention types mentioned in Section 5.2, but it's not clear to me which of those is the intervention type that is used when comparing observation-only versus observation with intervention in the approach. I'm assuming it's the variant that's just titled intervention, but I'm not sure.
- Significance
    - Weakness: Perhaps this is my bias toward more realistic tasks, but I don't feel like this paper meaningfully informed my understanding of model capabilities and the bottlenecks to accomplishing productive work in the real world.

**Weaknesses:**

See Strengths section (listed them together)

**Questions:**

1. Am I correct that only three attempts per task variant were performed?
2. Can you provide more details on how to interpret the descriptiveness score?
3. Why was the LLM limited to reasoning only once every five queries? Were any experiments done to show that that would be the optimal amount? For example, do LLMs get confused if they reason more often?

---

> ### Author Response · Authors · 2025-11-24
>
> We thank the reviewer for acknowledging our contribution in analyzing the fundamental cognitive components underlying AI scientist systems. We provide the response to the individual points below.
>
>  ## Originality
> > it doesn't seem to me that either the tasks or the methodology used in this paper represent meaningfully novel contributions.
>
> We agree that our tasks (programs, languages, and math equations) and methods (inductive reasoning and active learning) themselves are not novel. Our contribution is to use them as black-box systems and test whether LLMs can reliably reverse engineer them. To our knowledge, these tasks have not been used to evaluate LLMs this way. As the reviewer noted, our goal is to break down the basic reasoning steps an AI scientist needs, and we argue that reverse engineering --- generating hypotheses from observed data (inductive reasoning) and collecting data through interventions (active learning) --- captures key components needed for LLMs to automate scientific research. We have clarified this point in the paper.
>
> ## Quality
> > It seems like only three attempts per task were performed. This seems like a very low number of attempts. Granted, there are multiple “variants” of each task, corresponding to different numbers of data points, but only three runs on each task variant seems very low.
>
> We thank the reviewer for raising this important point about the number of evaluation attempts, which may not have been clear in the original paper. To clarify, in our paper, each type of black box includes many samples:
>
> - For Programs, we evaluate 100 instances × 3 seeds = 300 samples.
>
> - For Languages, we evaluate 46 instances × 3 seeds = 138 samples.
>
> - For Math Equations, we evaluate 50 instances × 3 seeds = 150 samples.
>
> In our original submission, we report error bars in Figure 2 and Figure 3 and provide repeated-measures ANOVA results in lines 268-270 and 298-300, which show that the current sample size is sufficient to support our conclusions.
>
> ## Re: weaknesses (second point in quality) & question 2
> > “interpret the descriptiveness score”
>
> The descriptiveness score is a rubric-based 0–10 metric rescaled to 0–100 in the paper.  We evaluate two things:
>
> (1) correctness: whether the inferred rule matches the underlying black-box behavior.
>
> (2) simplicity: whether it avoids unsupported or extraneous conditions.
>
> To show the qualitative difference, we provide two examples from the Formal Language tasks:
>
> Language rule: "AAAA"
>
> Language examples: "A, AA, AAA, AAAA"
>
> **Score of 30:**
>
> LM’s answer: All strings consist only of the character 'A', and the lengths follow the pattern of powers of 2.
>
> **Score of 50:**
>
> LM’s answer: The formal language generates all non-empty strings consisting only of the character 'A'. In other words, it generates A^n where n ≥ 1.
>
> As you can see, a score around 30 usually reflects only partial reconstruction, such as failing to generate some of the evaluation examples used for testing (e.g., missing "AAA" in the language task). A score around 50 indicates that the model has recovered the core structure while still making boundary-case errors or including minor unnecessary conditions. We have added these examples to the Appendix to make this clear.
>
> > The LLM transfer experiment feels distracting. I would not expect to see improved performance when transferring results from one model to another if we don't see improved performance within a model.
>
> The motivation for the transfer experiment comes from our earlier finding that GPT-4o benefits substantially from active intervention (Fig. 3). This naturally raises the question: if some models cannot formulate useful queries on their own, can they still benefit when given GPT-4o's intervention data? The experiment in Section 5.3 addresses this question directly.
> We agree that this analysis is not central to the main story, which is why we placed it in the analysis section rather than the main results. We are also happy to move it to the appendix in the final paper if necessary. The goal of this experiment is to show that the limitation of models is not only "asking the wrong questions," but that even when provided with relatively better data, they still fail to infer the underlying rule, pointing to a deeper reverse-engineering bottleneck.

---

> > ### Author Response · Authors · 2025-11-24
> >
> > ## Clarify
> > > The diagram at the start of the paper was not great; it was a bit busy and not terribly informative. It did not do much to improve my understanding of overcomplication and overlooking
> >
> > The teaser (Figure 1) is designed to 1) define the reverse-engineering problem; 2) show that LLMs can benefit from active interventions; and 3) provide examples of the two failure modes. We thank the reviewer for pointing out the lack of explanation for the failure examples. For overcomplication, the model adds an extra sorting step that is not supported by any (x, y) pairs when reversing a list-mapping program. For overlooking, the example shows the model provides the answer without reviewing the observed data when reversing a utility function. We’ve updated the caption for Figure 1 in the revised version.
> >
> > > it's not clear to me which of those is the intervention type that is used when comparing observation-only versus observation with intervention in the approach.
> >
> > We agree that we did not clearly specify which intervention variant was used. Our main comparison (Figures 2–4) uses the Analyze-then-Query intervention, and we have clarified this in the revised version.
> >
> > ## Significance
> > > I don't feel like this paper meaningfully informed my understanding of model capabilities and the bottlenecks to accomplishing productive work in the real world.
> >
> > We understand the reviewer's concern about realism. Our goal is to evaluate the reliability of LMs on the fundamental steps that real-world scientific discovery depends on, which requires a fully controllable setting. Such controlled tasks allow us to test whether the model can reliably use observations and interventions without confounds from task complexity or uncontrolled variables. Moreover, it is important to note that using cognitive science as a tool [1] in controlled, synthetic tasks represents a standard approach for exploring and understanding the challenges faced by LLMs [2, 3, 4]. This experimental setup provides the additional benefit of controllability, facilitating clean and interpretable LM evaluation.
> >
> > [1] Using the tools of cognitive science to understand large language models at different levels of analysis (2025)
> >
> > [2] Mind Your Step (by Step): Chain-of-Thought can Reduce Performance on Tasks where Thinking Makes Humans Worse (2025)
> >
> > [3] Cogbench: a large language model walks into a psychology lab. (2024)
> >
> > [4] Symbolic metaprogram search improves learning efficiency and explains rule learning in humans (2024)
> >
> >
> > ## Questions 3
> > > “Why was the LLM limited to reasoning only once every five queries? Were any experiments done to show that that would be the optimal amount? For example, do LLMs get confused if they reason more often?”
> >
> > The goal of the reverse-engineering task is to infer the underlying black-box rule from *all* observed data. In our early experiments, we tried prompting the model to reason after every query and found that the LM tended to focus too strongly on the current query–response pair. This is a hyperparameter that needs to be fixed early in the project, we set it to "reason every five queries."
> > To directly answer your question, we ran an additional comparison on the Language black-box using GPT-5 with \(n_{\text{obs}} = 10\) and \(n_{\text{intv}} = 10\). Reasoning every five queries yields a descriptive score of **46.7**, while reasoning after every single query produces a lower score of **40.9**.
> > To clarify, this choice is **not intended to be optimal**. It is a tunable hyperparameter, and we selected "every five queries" early in the project as a balance between performance, cost, and runtime across all experiments. We add the clarification in the revised paper in L. 431 (see footnote).

---

### Meta-Review · Area_Chair_Xw97 · 2026-01-06

**Summary:**

In "Assessing the Reverse-Engineering Abilities of Large Language Models", the authors test whether LLMs are able to reverse-engineer black-box systems, in the submitted paper exemplified by the synthetic tasks of 'list-mapping programs, formal languages, and inferring the parameters of an economics math equation'. The submission shows that existing models struggle with this task when provided with only passive observations of these systems, but improve when given the opportunity to actively query these systems.

**Reviewer Concerns:**

The main concern shared among reviewers concerns the design of the experiment and its evaluation protocol.

On the one hand, the submission uses a set of designs of clean-room, synthetic tasks (`mbzm`), such as list mapping and equation matching, which may raise concerns that these tasks are not realistic in relationship to the aim of testing the potential of LLMs to reverse-engineer black-box systems in science.

On the other hand (`sTYg`, `yYYx`), even though the tasks appear exactly verifiable, the submission uses only inexact model-based evaluation. While model-based evaluation is not never a problem, the reviewers were confused why it was employed here, and the authors assurance that the evaluation was (based on a limited human study) roughly correct, seemed not as convincing. During the rebuttal the authors argue with a number of studies that LLM-based evaluation can be good, but neglect a larger body of work showing problems with LLM-as-a-judge evaluations (such as self-bias raised as a concern by reviewer `xSfu`) showing the opposite. Instead of arguing that this evaluation protocol may be mostly correct, it appears unclear to the reviewers (and to me), why the authors do not implement fully verifiable evaluation.

Strangely, with these two issues, the evaluation protocol appears to combine the problems with evaluation of a real-world black-box system reverse-engineering (where the model-based scoring would be the only option), with its simple clean-room tasks (that could have been verified). The rebuttal indirectly defends this as a fact-of-life description of already conducted experiments, but this is a conscious design choice in their digital experimental design that the reviewers can criticize.


There are a number of smaller concerns raised during the review process, such 'performance over multiple seeds', relationship of number of observations to performance of LLM in active inference (or lack thereof), smaller writing and diagram concerns and questions regarding the performance of active inference when controlling for reasoning length (measured in number of tokens) that I feel like the author responses address adequately.

Overall though, given the shared concerns of the reviewers regarding the experimental design, I am not recommending acceptance of the currently submitted draft.

**Reviewer Scores:**

`mbzm` could have increased their score to a 6. `sTYg` remains slightly positive. `xSfu` and `yYYx` explicitly note during the discussion that they are unconvinced by the author response.

---

### Decision · Program_Chairs · 2026-01-26

Reject